# Differential regulation of coronal and lambdoid suture patency by PTHLH and HHIP activity in mice

Madrikha D. Saturne[1,*], Susan M. Motch Perrine[2,‡], Qingyang Li[1], Joan T. Richtsmeier[2], Ethylin Wang Jabs[1,§], Harm van Bakel[1,3,4,5] and Greg Holmes[1,¶]

## ABSTRACT

Craniofacial development depends on the formation of fibrous joints, or sutures, between skull bones. Premature fusion of sutures, or craniosynostosis, is a common human pathology. Ectopic Hedgehog (HH) signaling is one cause of craniosynostosis. *Hhip* encodes an inhibitor of HH ligands, and we previously identified coronal suture dysgenesis in embryonic *Hhip*$^{-/-}$ mice, in which suture mesenchyme was depleted between closely opposed but unfused osteogenic fronts at E18.5. Here, we report that the lambdoid suture fuses in *Hhip*$^{-/-}$ mice by E18.5. RNA-seq analysis of the *Hhip*$^{-/-}$ coronal and lambdoid sutures show that HH target gene expression, including *Pthlh*, is upregulated. Paradoxically, expression of *Ihh* is downregulated. We hypothesized that PTHLH, a negative regulator of *Ihh* expression, may reduce HH signaling to promote coronal suture patency and prevent fusion of the *Hhip*$^{-/-}$ coronal suture. We generated *Hhip*$^{-/-}$;*Pthlh*$^{-/-}$ embryos and found that coronal sutures are fusing by E18.5. Our results reveal a previously undescribed role for *Pthlh* in suture development and demonstrate suture-specific roles for HH inhibitors in maintaining suture patency.

KEY WORDS: Suture, Hedgehog, Parathyroid, Craniosynostosis, Coronal, Lambdoid

## INTRODUCTION

Development of the craniofacial skeleton depends on proper formation of fibrous joints, or sutures, between skull bones (Ishii et al., 2015). Sutures consist of the edges of adjacent craniofacial bones and the intervening suture mesenchyme (SM). Growth occurs at the bone edges, or osteogenic fronts (OFs), by intramembranous ossification, whereby progenitor cells differentiate directly to

[1]Department of Genetics and Genomic Sciences, Icahn School of Medicine at Mount Sinai, New York, NY 10029, USA. [2]Department of Anthropology, Pennsylvania State University, University Park, PA 16802, USA. [3]Icahn Genomics Institute, Icahn School of Medicine at Mount Sinai, New York, NY 10029, USA. [4]Department of Microbiology, Icahn School of Medicine at Mount Sinai, New York, NY 10029, USA. [5]Windreich Department of Artificial Intelligence and Human Health, Icahn School of Medicine at Mount Sinai, New York, NY 10029, USA.
*Present address: Department of Genetics and Development, Columbia University Irving Medical Center, New York, NY 10032, USA. ‡Present address: Department of Medicine, College of Medicine, Pennsylvania State University, Hershey, PA 17033, USA. §Present address: Department of Clinical Genomics and Department of Biochemistry and Molecular Biology, Mayo Clinic, Rochester, MN 55905, USA.

¶Author for correspondence (gregory.holmes@mssm.edu)

M.D.S., 0000-0001-6677-5675; S.M.M.P., 0000-0003-3412-221X; J.T.R., 0000-0002-0239-5822; E.W.J., 0000-0001-8983-5466; H.v.B., 0000-0002-1376-6916; G.H., 0000-0003-0717-6722

osteoblasts. The SM separates craniofacial bones, and postnatally forms a niche for suture stem cells required to maintain and repair bone (Debnath et al., 2018; Maruyama et al., 2016; Wilk et al., 2017; Zhao et al., 2015). Many signaling pathways regulate these processes (Ishii et al., 2015).

Hedgehog (HH) signaling is one such pathway. HH signaling occurs when ligands such as sonic hedgehog (SHH) and Indian hedgehog (IHH) bind patched receptors (PTCH1 and PTCH2) to relieve PTCH inhibition of the HH signaling effector smoothened (SMO) and upregulate expression of HH target genes. These include *Gli1*, which encodes a transcriptional activator of HH target genes, and *Ptch1* and *Hhip*, whose proteins inhibit signaling by sequestering HH ligands in a negative-feedback loop (Lee et al., 2016). IHH is expressed in the OFs (Jacob et al., 2007; Rice et al., 2010), where it promotes osteogenesis (Klopocki et al., 2011; Lenton et al., 2011; Veistinen et al., 2017) and is the functional HH ligand in suture development (Veistinen et al., 2017; Zhao et al., 2015). Loss of HH signaling in murine *Ihh*$^{-/-}$ calvaria, or the skull roof, reduces ossification, resulting in wider sutures (Abzhanov et al., 2007; Klopocki et al., 2011; Lenton et al., 2011; Veistinen et al., 2017). IHH upregulates genes such as *Bmp2/4* and *Igf2* to promote the expression and activity of *Runx2*/RUNX2, the master regulator of osteogenesis (Lenton et al., 2011; Shi et al., 2015). RUNX2, in turn, upregulates expression of *Ihh* and other pro-osteogenic genes (Qin et al., 2019). Postnatally, HH signaling regulates the differentiation of suture stem cells (Guo et al., 2018; Zhao et al., 2015).

Craniosynostosis (CS) occurs when one or more craniofacial sutures fuse prematurely. Suture fusion in the calvaria can lead to distorted skull growth and neurological defects due to increased intracranial pressure (Stanton et al., 2022). CS is one of the most common craniofacial defects in humans and is caused by mutations in a wide variety of genes (Heuzé et al., 2014; Ishii et al., 2015; Twigg and Wilkie, 2015). The human calvaria consists of paired frontal bones, paired parietal bones and the occipital bone (or murine interparietal bone), arranged anteroposteriorly. The coronal suture, between the frontal and parietal bones, is the second-most affected suture in non-syndromic CS but is the most commonly affected suture in syndromic CS. Fusion of the lambdoid suture, between the parietal and occipital bones, is a rare event (Heuzé et al., 2014; Wilkie et al., 2017).

Mutations that enhance HH signaling can cause CS. In loss-of-function mouse mutants of *Gli3*, a HH signaling repressor, HH target genes are upregulated and the lambdoid suture fuses (Rice et al., 2010). Excess HH signaling causes lambdoid suture fusion in a *Ptch1* hypomorph (Feng et al., 2013). In mice and humans, duplications of *Ihh/IHH* regulatory elements cause CS affecting various sutures, including the coronal suture (Barroso et al., 2015; Klopocki et al., 2011; Will et al., 2017). Activating mutations of *SMO* cause Curry Jones syndrome, in which coronal sutures fuse (Twigg et al., 2016). Loss-of-function mutations of *RAB23*, a HH inhibitor, cause

Carpenter syndrome, in which the coronal and other sutures fuse (Jenkins et al., 2007). Loss-of-function mutations of *GLI3* cause metopic and sagittal suture fusion in Greig cephalopolysyndactyly syndrome (Hurst et al., 2011; McDonald-McGinn et al., 2010).

It is evident that negative regulation of HH signaling is important for proper morphogenesis, but how this occurs in sutures is poorly understood. Increased levels of PTCH1 in response to HH signaling reduce the diffusion and increase the internalization of HH ligand (Lee et al., 2016). HHIP binds to HH ligand to reduce its diffusion and availability for binding to PTCH1, further reducing HH signaling (Lee et al., 2016). We have previously shown that *Hhip* expression is enriched in coronal SM and coronal suture formation is defective in embryonic day (E) 18.5 *Hhip*$^{-/-}$ embryos, with loss of the overlapping structure of the frontal and parietal bones and depletion of intervening SM, but without fusion of the bones (Holmes et al., 2021). Here, we show that by E18.5 *Hhip*$^{-/-}$ embryos have fusion of the lambdoid suture. Surprisingly, despite increased HH signaling, *Ihh* expression was decreased in mutant lambdoid and coronal sutures. We characterized transcriptional changes in both mutant sutures by RNA-seq and identified *Pthlh* as a potential negative regulator of *Ihh* transcription. We generated *Hhip*$^{-/-}$;*Pthlh*$^{-/-}$ embryos and found that by E18.5 the coronal suture fused, a stronger phenotype than that of *Hhip*$^{-/-}$ coronal sutures. Our study identifies a previously unreported interaction between *Ihh* and *Pthlh* in sutures, reminiscent of their interaction in long bone growth plates (Ohba, 2020). We demonstrate the critical importance of negative regulation of the HH pathway at both the protein and transcription levels of IHH for normal suture development and the prevention of CS.

## RESULTS
### Loss of *Hhip* increases skull size and calvarial bone thickness
We have previously described coronal suture dysgenesis, with the loss of SM and close approximation of frontal and parietal bones, in E18.5 *Hhip*$^{-/-}$ embryos (Holmes et al., 2021). Here, we further investigated changes in the skull of *Hhip*$^{-/-}$ embryos. Microcomputed tomography (microCT) imaging at E18.5 showed a clear distinction between wild-type and *Hhip*$^{-/-}$ skulls (*n*=11 and 10, respectively). We performed a Generalized Procrustes Analysis (GPA) superimposition of the coordinate data of 39 anatomical landmarks (Table S1) describing the facial skeleton, cranial vault and cranial base of wild-type and *Hhip*$^{-/-}$ embryos to extract shape information. Principal components analysis (PCA) of GPA-aligned shape coordinates of landmark data showed a clear separation of genotypes on PC1, indicating a size difference (allometry; Fig. 1A), with *Hhip*$^{-/-}$ skulls being larger. After correction for allometry, separation of genotypes remained along PC1 and PC2, indicating subtle shape differences (Fig. 1B).

Within genotypes, the paired frontal and parietal bone volumes were similar (Fig. 1C). Between genotypes, *Hhip*$^{-/-}$ frontal and parietal bone volumes were larger than wild type, with no difference in interparietal volumes (Fig. 1C). Frontal and parietal bone thickness ranged from just over 0 μm to just under 70 μm (Fig. 1D-G). The orbital plates of both genotypes displayed the greatest thickness, with *Hhip*$^{-/-}$ embryos showing an overall increased bone thickness compared to wild type (Fig. 1F,G). The mutant areas of greatest bone thickness (red in Fig. 1F,G), as determined by increased pixel density, also mapped to areas that showed increased bone mineral density, as determined by higher pixel intensity along the calibration curve defined by a co-scanned hydroxyapatite phantom. Euclidean Distance Matrix Analysis (EDMA) of the landmark data indicated an overall enlargement of the *Hhip*$^{-/-}$ skull, with all linear distances

between landmarks shown representing a significant 5-17% increase (Fig. 1H,I). Linear distances that define the cranial vault area encompassing the frontal and parietal bones showed the largest increase (10-17%), with the remaining measures indicating an increase of 5-10% in *Hhip*$^{-/-}$ specimens.

### Loss of *Hhip* causes lambdoid suture fusion
For all sutures compared by microCT imaging, there was no significant difference in the degree of suture fusion, except for the right nasal-premaxillary suture (*P*=0.035) in which the *Hhip*$^{-/-}$ embryos had less fusion than wild type (Table S2). Three of 10 *Hhip*$^{-/-}$ embryos had partial fusion of the lambdoid suture, in contrast to patent lambdoid sutures in the wild type, indicating a trend towards fusion, though not significant (*P*=0.056) (Fig. 1J,K and Table S2). Ectopic bone at the lower posterior margin of the parietal bone was seen in all *Hhip*$^{-/-}$ embryos, with 90% of cases being bilateral (Fig. 1J,K).

Histological sectioning allows visualization of non-mineralized bone matrix (osteoid) not imaged by microCT. Fusion, or continuous osteoid between the parietal and interparietal bones, was identified in lambdoid sutures of all E18.5 *Hhip*$^{-/-}$embryos examined (Fig. 2; *n*=10). Unlike the coronal suture, the parietal and interparietal bones of the lambdoid suture are arranged end-to-end, rather than overlapping, during embryonic development. At E18.5, alkaline phosphatase (ALP) staining of the wild-type suture demonstrated a relatively wide SM separating parietal and interparietal bones (Fig. 2A). In *Hhip*$^{-/-}$ embryos, there was fusion within the suture bilaterally (7/10) or unilaterally (3/10), with a closer approximation of the bone edges near points of fusion (Fig. 2A). No fusion was seen in wild-type embryos (7/7). RUNX2 immunohistochemistry of the wild-type lambdoid suture showed expression throughout the SM between the calvarial bones, with decreasing intensity away from the OFs (Fig. 2B). SP7 expression, a marker of preosteoblasts and osteoblasts, was restricted to the parietal and interparietal bones, similar to ALP activity (Fig. 2C). The relative location of RUNX2 and SP7 expression did not change in the mutant sutures, except where continuous RUNX2 and SP7 expression occurred in fused regions (Fig. 2B,C). Changes in cell proliferation were determined by quantifying EdU incorporation as a percentage of total cell number in individual parietal and interparietal OFs and SM. Proliferation in the OFs and SM did not differ between wild-type and unfused regions of *Hhip*$^{-/-}$ lambdoid sutures (Fig. 2D,E). There was a significant decrease in total cell number in the *Hhip*$^{-/-}$ SM, reflecting a shortening of the mutant SM near fused regions, and a small but significant increase in cell numbers in the interparietal OF (Fig. 2F; *t*-test, *P*=0.033 and 0.042, respectively; *n*=6 wild type and 6 *Hhip*$^{-/-}$). No apoptotic cells were present in either the wild-type or *Hhip*$^{-/-}$ lambdoid sutures (Fig. S1). As in the *Hhip*$^{-/-}$ coronal suture (Holmes et al., 2021), loss of IHH inhibition advanced ossification from the OFs with loss of SM. However, the *Hhip*$^{-/-}$ lambdoid suture phenotype is more severe than the coronal, which remained unfused by E18.5 (Holmes et al., 2021).

We assessed the patency of *Hhip*$^{-/-}$ facial sutures at E18.5 in histological sections. No fusion was seen in major sutures such as the internasal, interpremaxillary, intermaxillary, interpalatine and maxillary-palatine (Fig. S2), in agreement with microCT imaging (Table S2).

### *Ihh* expression decreases in the *Hhip*$^{-/-}$ lambdoid and coronal sutures, despite increased HH signaling
To understand why the lambdoid suture phenotype is stronger than the coronal, we first investigated differences in the expression of *Ihh*

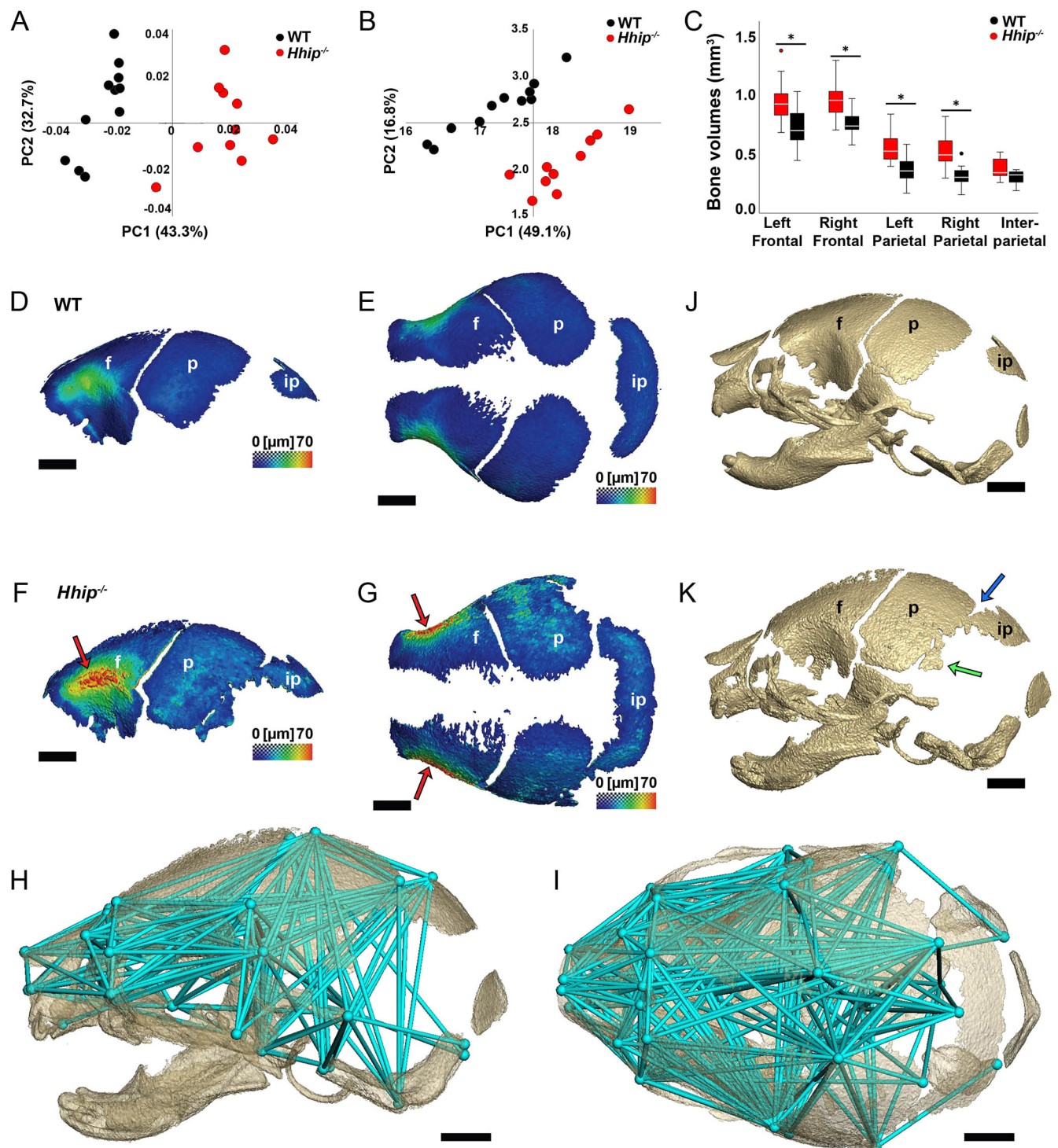

**Fig. 1. *Hhip* deletion causes increased calvarial size and bone thickness, and ectopic bone formation by E18.5.** (A,B) PCA of skull morphology using the 3D coordinates of 39 anatomical landmarks. Results are for wild-type (*n*=11) and *Hhip*−/− (*n*=10) skulls at E18.5 to demonstrate differences between groups in (A) overall form without scaling and (B) shape (adjusting for allometry). (C) Individual calvarial bone volumes. Boxes indicate the interquartile range and median; whiskers indicate the non-outlier maximum and minimum values. Asterisks indicate significant differences. *P*-values, from left to right: 0.024, 0.006, 0.004 and 0.003. (D-K) 3D reconstructions of microCT images thresholded for bone at E18.5. (D-G) Thickness maps of frontal (f), parietal (p) and interparietal (ip) bones in representative lateral (D,F) and superior (E,G) views of wild-type (D,E) and *Hhip*−/− (F,G) calvaria. Colormap scale indicates bone thickness from 0 to 70 µm. Red arrows (F,G) indicate increased thickness of orbital plates. (H,I) Lateral (H) and superior (I) views of EDMA results displaying 234 unique interlandmark linear distances (aqua lines) that are significantly larger in *Hhip*−/− mice. Aqua spheres represent anatomical landmarks and linear distance endpoints. (J,K) Lateral views of representative (J) wild-type and (K) *Hhip*−/− skulls. Blue arrow indicates mineralized fusion of lambdoid suture. Green arrow indicates ectopic parietal bone. Scale bars: 1 mm.

and three transcriptional readouts of HH signaling, *Gli1*, *Ptch1* and *Hhip*, by multiplexed single molecule fluorescence *in situ* hybridization (smFISH). In the wild-type E18.5 lambdoid suture, *Ihh* was expressed in the OFs; *Gli1* and *Ptch1* were expressed in the OFs and adjacent SM (Fig. 3A). This was similar to their expression pattern in the coronal suture (Fig. 3D) (Holmes et al., 2021). *Hhip*

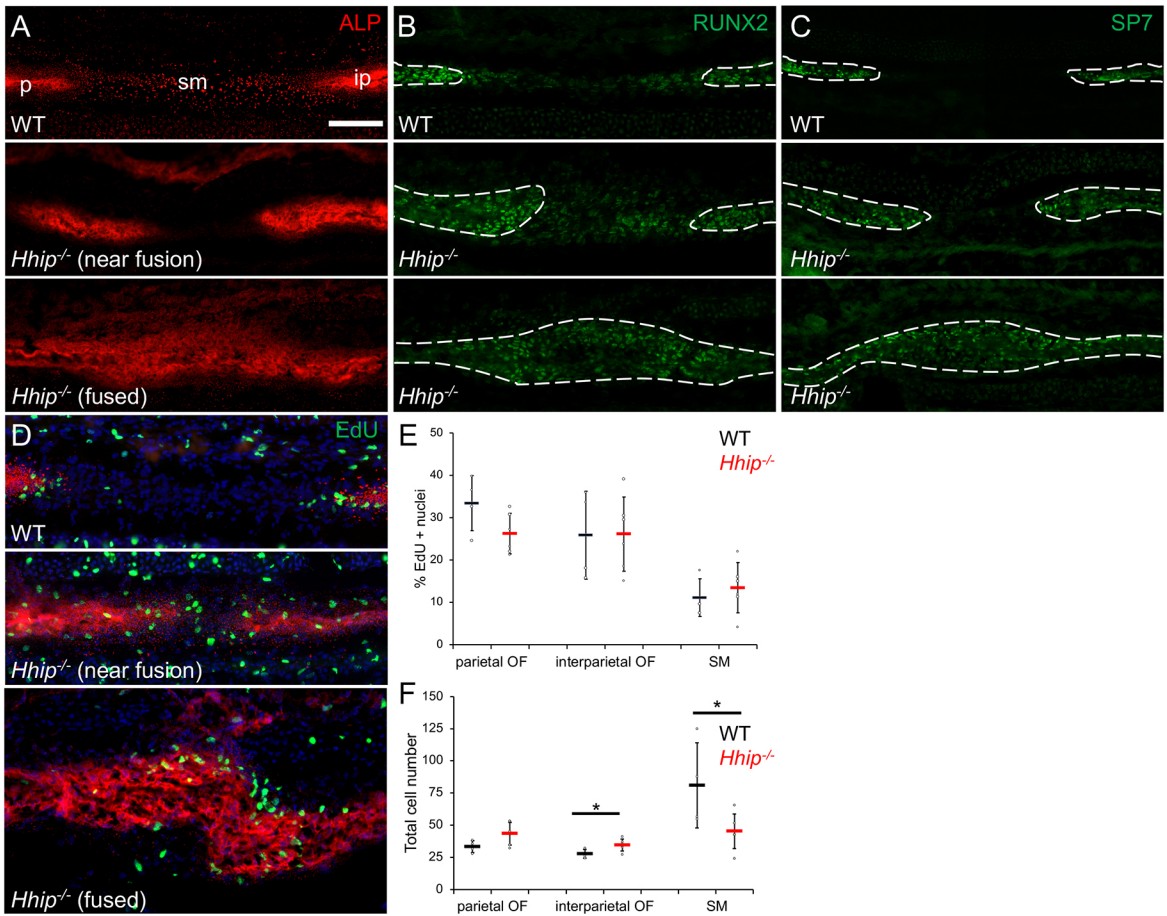

**Fig. 2. *Hhip*⁻ᐟ⁻ lambdoid sutures fuse by E18.5.** (A) Alkaline phosphatase activity (ALP, red), (B) RUNX2 immunohistochemistry (green) and (C) SP7 immunohistochemistry (green) of wild-type (top; *n*=7 in A and 3 in B,C) and *Hhip*⁻/⁻ (*n*=10 in A and 3 in B,C) near fusion (middle) and fused (bottom) lambdoid sutures. (D) EdU incorporation assay (green), counterstained for ALP activity (red) and DAPI (blue). (E) Quantification of EdU-positive nuclei in OFs and SM of wild-type (black) and *Hhip*⁻/⁻ (near fusion; red) sutures. (F) Quantification of total cells in indicated OFs and SM (unpaired *t*-test, *P*=0.033 and 0.042, respectively; *n*=6 wild type and 6 *Hhip*⁻/⁻ in D-F). White dashed lines indicate interparietal (ip) and parietal (p) bones. sm, suture mesenchyme. Sections are in the transverse plane. Scale bar: 50 µm.

was expressed in the OFs and adjacent SM, similar to *Gli1* and *Ptch1* (Fig. 3A). This differed from the coronal suture, where expression extends from the OFs but is highest throughout the SM between the overlapping frontal and parietal bones (Fig. 3D) (Holmes et al., 2021). Surprisingly, in the *Hhip*⁻/⁻ lambdoid suture the domain and intensity of *Ihh* expression clearly was decreased in the OFs of unfused regions (Fig. 3B), while *Gli1* and *Ptch1* expression was higher throughout the SM (Fig. 3B). In fused regions of *Hhip*⁻/⁻ lambdoid sutures, *Ihh* expression was high in osteoblasts, and *Gli1* and *Ptch1* expression was widespread in ecto- and endocranial mesenchyme adjacent to the fusion point (Fig. 3C). The domain and intensity of *Ihh* expression also was strongly decreased in the OFs of the *Hhip*⁻/⁻ coronal suture (Fig. 3E).

The *Hhip*⁻/⁻ mouse incorporates a LacZ reporter driven by the endogenous *Hhip* promoter. As *Hhip* is a target of HH signaling, LacZ expression provides a readout of HH activity at the *Hhip* locus (Chuang and McMahon, 1999). *Hhip*⁻/⁻ embryos contain two copies of this reporter, while *Hhip*⁺/⁻ embryos contain one copy. *Hhip*⁻/⁻ lambdoid sutures exhibited disproportionately higher LacZ expression compared to *Hhip*⁺/⁻ sutures at E18.5, which extended along bone surfaces beyond the suture (Fig. 3F). A similar disproportionate increase in LacZ expression was seen between the *Hhip*⁺/⁻ and *Hhip*⁻/⁻ coronal sutures (Fig. 3G), as we had reported previously (Holmes et al., 2021). LacZ expression was also typically

stronger in the *Hhip*⁺/⁻ coronal suture compared to the lambdoid suture (Fig. 3F,G), in agreement with *Hhip* expression levels when determined by smFISH (Fig. 3A,D) and RNA-seq (Table 1; see below). Taken together, these results show that, in both sutures, HH transcriptional outputs are increased throughout the *Hhip*⁻/⁻ SM compared to wild type, even as *Ihh* expression in the OFs is decreased.

## Transcriptome analysis of *Hhip*⁻/⁻ coronal suture defects

Reduction of *Ihh* expression in *Hhip*⁻/⁻ coronal and lambdoid sutures was unexpected. For example, in the *Hhip*⁻/⁻ embryo, no reduction in expression of *Shh* in the lung or of *Ihh* in the vertebral cartilage was noted (Chuang et al., 2003). This suggests a previously unidentified mechanism of *Ihh* regulation common to both sutures. The less severe coronal suture phenotype was also unexpected, given the prominence of *Hhip* expression between the frontal and parietal bones between E16.5 and E18.5 (Fig. 3D) (Holmes et al., 2021). This, and the lack of clear phenotypes in other sutures, suggests that suture-specific transcriptional environments account for such differences. To comprehensively identify and compare the transcriptional consequences of loss of *Hhip* within coronal and lambdoid sutures, we performed RNA-seq analysis of *Hhip*⁻/⁻ and wild-type sutures at E18.5.

Between *Hhip*⁻/⁻ and wild-type coronal sutures we identified 1156 differentially expressed genes (DEGs), of which 655 were

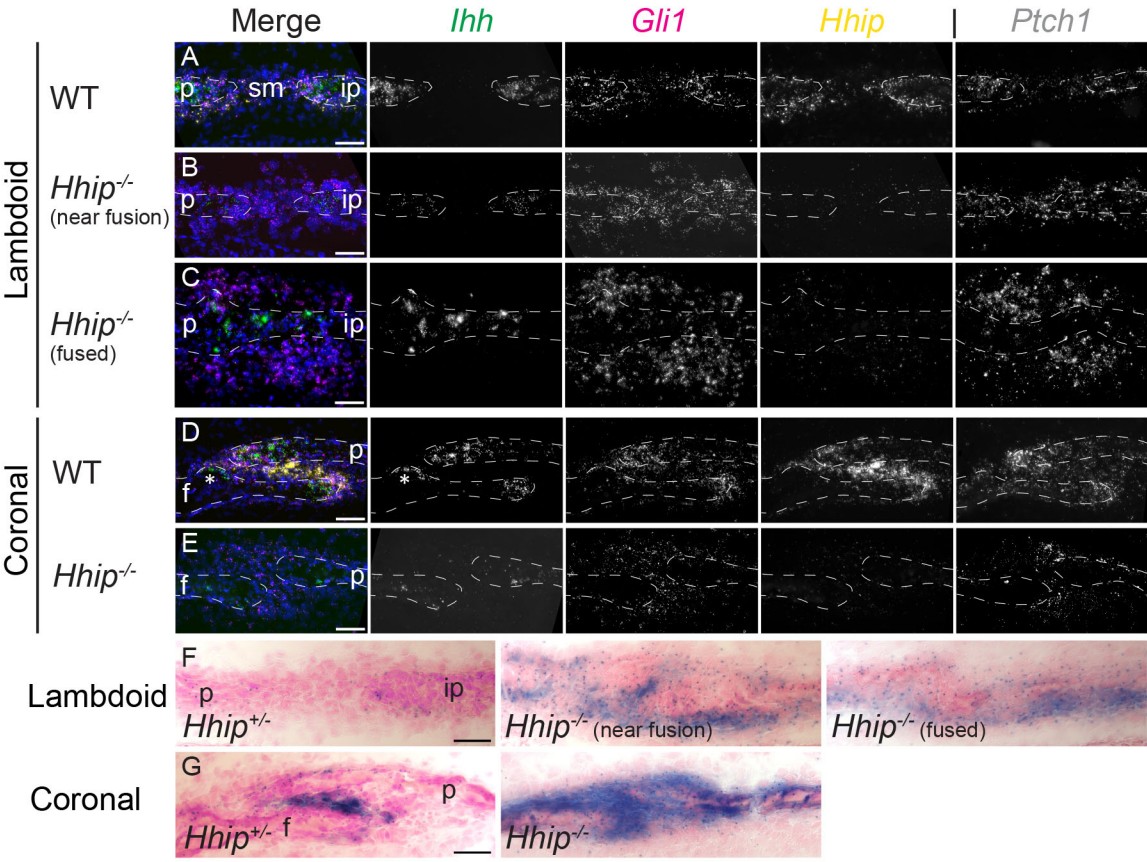

**Fig. 3. *Ihh* expression is decreased in osteogenic fronts but HH signaling is increased throughout the suture mesenchyme in E18.5 *Hhip*⁻/⁻ lambdoid and coronal sutures.** (A-E) smFISH for co-expression of the indicated HH pathway genes in lambdoid (A-C) and coronal (D,E) sutures. (A) Wild-type, (B) *Hhip*⁻/⁻ near fusion and (C) *Hhip*⁻/⁻ fused region of lambdoid suture. (D) Wild-type and (E) *Hhip*⁻/⁻ coronal suture. Gene colors in merged images are indicated by text color above greyscale images. *Ihh*, *Gli1*, *Hhip* and *Ptch1* expression are from adjacent or near-adjacent sections. (F,G) LacZ staining (blue) of *Hhip*⁺/⁻ and *Hhip*⁻/⁻ lambdoid (F) and coronal (G) sutures, counterstained with nuclear Fast Red. White dashed lines in A-E indicate frontal (f), interparietal (ip) and parietal (p) bones. Asterisks in D indicate localized *Ihh* expression in frontal bone opposite the parietal osteogenic front typically seen in lower suture area. For A-E, *n*=3 wild type and 3 *Hhip*⁻/⁻; for F,G, *n*=2 *Hhip*⁺/⁻ and *n*=3 *Hhip*⁻/⁻. Sections are in the transverse plane. Scale bars: 50 μm.

upregulated and 501 were downregulated in the *Hhip*⁻/⁻ suture compared to wild type (false discovery rate [FDR]<0.05) (Fig. 4A; Table S3). By Gene Ontology (GO) analysis, upregulated genes were enriched for Biological Process (BP) terms related to regulation of motility and migration (Table S3). The most enriched Kyoto Encyclopedia of Genes and Genomes (KEGG) term was 'proteoglycans in cancer' (Table S3). Downregulated genes were enriched for BP terms related to 'synapse' or 'cytoplasmic translation', driven by decreased expression of large and small ribosomal genes, and 'bone mineralization', 'bone remodeling' and 'osteoblast differentiation' (Table S3). The most enriched KEGG terms were for 'lysosome', driven by decreased expression of osteoclast markers, and 'ribosome', driven by decreased expression of large and small ribosomal genes (Table S3).

As increased IHH signaling might be expected to increase bone mineralization, we reviewed the downregulated genes within the GO BP term 'bone mineralization'. The bone markers included *Bglap, Bglap2, Ibsp* and *Ifitm5*, which were expressed at 48-71% of wild-type levels (Table S3). No significant expression changes were seen for the osteoblast fate determinant genes *Runx2* and *Sp7*, the general osteoblast marker *Alpl*, or the mature osteoblast marker *Spp1* (Table S3). The downregulation of these specific bone marker genes may therefore indicate a subtle shift in the relative distribution of osteoblast maturation states between wild-type and mutant at or adjacent to the coronal suture OFs.

Previously, we found by smFISH that expression of HH target genes *Gli1* and *Ptch1* were upregulated throughout *Hhip*⁻/⁻ coronal SM (Fig. 3E) (Holmes et al., 2021). The RNA-seq data showed that expression of *Gli1* and *Ptch1* was significantly upregulated in *Hhip*⁻/⁻ coronal sutures (log2 FC=0.50 and 0.59, respectively; Fig. 4E, Table 1 and Table S3). *Ihh* expression was significantly downregulated in *Hhip*⁻/⁻ sutures (log2 FC=−1.08), consistent with smFISH results (Fig. 3E).

We also assessed expression changes for genes known to positively or negatively regulate *Ihh* expression in chondrocytes or other cell types (Fig. 4E; Table 1 and Table S3). Among positive regulators, no significant change was seen for *Runx2* (Yoshida et al., 2004), *Msx2* (Amano et al., 2008) or *Sirt6* (Piao et al., 2013); *Atf4* (Wang et al., 2009) was mildly upregulated, while *Lef1* (Später et al., 2006) was downregulated. Among negative regulators, *Hand1* (Laurie et al., 2016) was not found to be expressed, and *Zeb1* (Bellon et al., 2009) expression was unchanged. In contrast, *Pthlh* (St-Jacques et al., 1999; Vortkamp et al., 1996) expression was upregulated in *Hhip*⁻/⁻ coronal sutures (log2 FC=0.82; Fig. 4E; Table 1 and Table S3). *Pthlh* and *Ihh* form a regulatory loop controlling the cartilage growth plate of long bones (Ohba, 2020; St-Jacques et al., 1999; Vortkamp et al., 1996), but a role for *Pthlh* in sutures has not been characterized. Expression of the gene encoding the receptor for PTHLH, *Pth1r*, was slightly but significantly downregulated (log2 FC=−0.29; Table S3).

**Table 1. Differential expression of Hedgehog pathway components**

| | Coronal $Hhip^{-/-}$ versus wild type | | Lambdoid $Hhip^{-/-}$ versus wild type | | Wild-type coronal versus lambdoid | |
|---|---|---|---|---|---|---|
| | log2FC | adj. $P$-value* | log2FC | adj. $P$-value* | log2FC | adj. $P$-value* |
| HH pathway genes | | | | | | |
| Hhip | −5.45 | 2.83E-11 | −5.20 | 6.43E-09 | 1.04 | 9.76E-07 |
| Gli1 | 0.50 | 0.01 | 0.36 | **0.05** | −0.10 | 0.53 |
| Gli2 | 0.28 | 0.13 | 0.01 | 0.99 | 0.04 | 0.85 |
| Gli3 | 0.13 | 0.47 | −0.12 | 0.54 | −0.15 | 0.28 |
| Ptch1 | 0.59 | 2.94E-03 | 0.63 | 1.54E-03 | 0.01 | 0.97 |
| Ptch2 | 1.36 | 3.95E-04 | 1.74 | 6.18E-05 | 0.43 | 0.24 |
| Ptchd4 | −0.59 | 0.46 | −0.07 | 0.93 | −1.33 | 0.01 |
| Disp1 | −0.25 | 0.13 | −0.26 | 0.12 | −0.29 | 0.03 |
| Ihh | −1.08 | 2.20E-04 | −1.08 | 7.28E-05 | −0.23 | 0.18 |
| Cdon | 0.41 | **0.05** | −0.39 | 0.07 | −0.65 | 1.04E-03 |
| Boc | −0.12 | 0.38 | −0.13 | 0.40 | 0.11 | 0.36 |
| Gas1 | −0.39 | 4.03E-03 | −0.26 | 0.04 | −0.25 | 0.02 |
| Smo | −0.03 | 0.76 | −0.04 | 0.74 | −0.17 | 0.02 |
| Sufu | 0.00 | 0.98 | −0.12 | 0.47 | −0.04 | 0.75 |
| Scube1 | 0.48 | 0.01 | 0.95 | 1.20E-03 | 1.64 | 9.59E-07 |
| Scube2 | −0.11 | 0.76 | −0.01 | 0.98 | −0.49 | 0.03 |
| Scube3 | −0.23 | 0.30 | −0.32 | 0.12 | −0.50 | 0.01 |
| Rab23 | 0.33 | 0.04 | −0.48 | 0.01 | 0.16 | 0.27 |
| Positive regulators of Ihh transcription | | | | | | |
| Runx2 | −0.16 | 0.23 | −0.17 | 0.29 | −0.11 | 0.37 |
| Msx2 | −0.31 | 0.21 | −0.37 | **0.05** | −0.46 | 0.01 |
| Sirt6 | −0.07 | 0.77 | −0.16 | 0.33 | −0.16 | 0.20 |
| Atf4 | 0.32 | 0.03 | 0.01 | 0.96 | 0.01 | 0.96 |
| Lef1 | −1.24 | 0.01 | −0.45 | 0.41 | 0.21 | 0.58 |
| Negative regulators of Ihh transcription | | | | | | |
| Zeb1 | 0.17 | 0.17 | −0.03 | 0.85 | −0.15 | 0.12 |
| Pthlh | 0.82 | 1.56E-03 | 2.56 | 2.68E-03 | 4.70 | 1.14E-06 |

*$P<0.05$ are shaded in gray; $P=0.05$ are in bold.

Various genes associated with craniofacial or bone phenotypes were found among the DEGs with the highest fold change. Genes upregulated in the mutant include *Fosl1*, *Ly6a*, *Wnt4*, *Sfn*, *Foxf2*, *Tll1* and *Bnc2*. Genes downregulated in the mutant include *Hmx1*, *Grem1*, *Nell1*, *Barx1*, *Fgf10*, *Fgf10* and *Ihh* (Table S3).

**Transcriptome analysis of $Hhip^{-/-}$ lambdoid suture defects**
Between $Hhip^{-/-}$ and wild-type lambdoid sutures, we identified 1321 DEGs, of which 836 were upregulated and 485 were downregulated in the $Hhip^{-/-}$ suture compared to wild type (FDR<0.05) (Fig. 4A; Table S3). By GO analysis, upregulated genes were enriched for a large number of BP terms, including those related to cell migration, 'blood vessel development', 'osteoclast differentiation' and 'biomineral tissue development' (Table S3). The most enriched KEGG term was 'osteoclast differentiation' (Table S3). Downregulated genes were enriched for GO BP terms related to 'nervous system development' and 'skeletal system development' (Table S3). The only enriched KEGG term was 'axon guidance' (Table S3).

We examined the expression of HH pathway components in the lambdoid suture. HH target genes *Gli1* and *Ptch1* were upregulated

in the $Hhip^{-/-}$ suture (log2 FC=0.36 and 0.63, respectively; Fig. 4E; Table 1 and Table S3), although *Gli1* upregulation was borderline significant. Again, *Ihh* expression was significantly downregulated in $Hhip^{-/-}$ sutures (log2 FC=−1.08). These changes were consistent with smFISH results (Fig. 3).

Most regulators of *Ihh* expression were not differentially expressed. Expression of the positive regulator *Msx2* was downregulated, but significance was borderline (Table 1 and Table S3). As in the coronal suture, *Pthlh* expression was upregulated significantly in $Hhip^{-/-}$ sutures (log2 FC=2.56; Fig. 4E; Table 1 and Table S3).

Of the lambdoid DEGs, many of the most highly upregulated genes in the mutant were osteoclast markers (e.g. 9 of the top 15), and included *Atp6v0d2*, *Siglec15*, *Ccr3*, *Mmp9*, *Gpr55*, *Ocstamp* and *Dcstamp* (Table S3). Mutation of many of these genes in the mouse results in craniofacial and/or bone phenotypes due to impaired osteoclast remodeling of bone. Notably, inducers of osteoclast differentiation, such as *Tnfsf11* (*Rankl*) and *Csf1*, were upregulated (Table S3). The genes most highly downregulated and whose mutation in mouse results in craniofacial phenotypes included *Grm3*, *Nell1*, *Pou3f3*, *Slc13a4*, *Rspo2*, *Wnt10a* and *Dmrt2*.

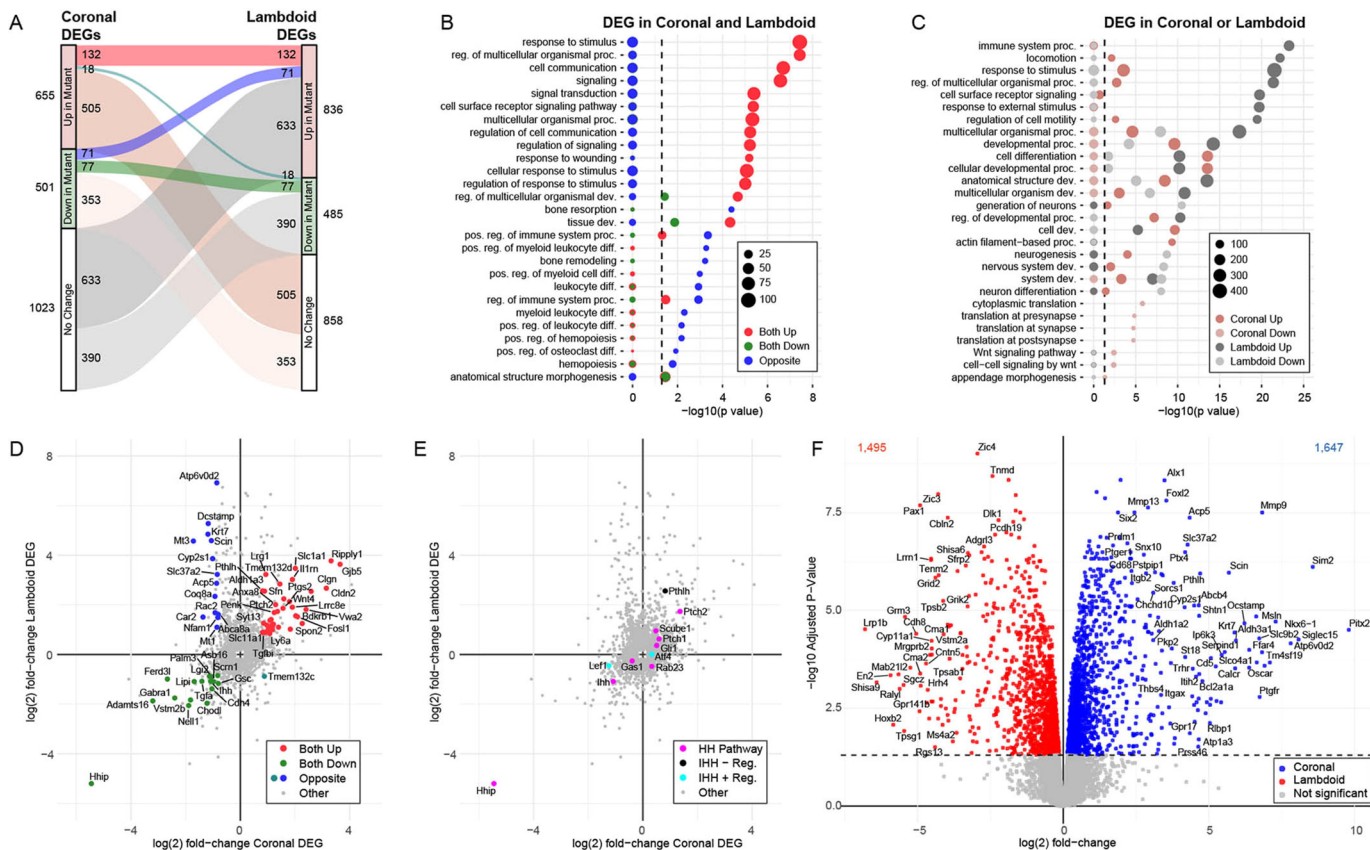

**Fig. 4.** *Hhip* **deletion differentially alters gene expression in E18.5 coronal and lambdoid sutures.** (A) Alluvial plot comparing DEGs between wild-type and *Hhip*⁻/⁻ coronal (left; *n*=4 wild type and 3 *Hhip*⁻/⁻) and lambdoid (right; *n*=4 wild type and 4 *Hhip*⁻/⁻) sutures at E18.5. Significant DEGs in both sutures with the same or opposite expression changes are highlighted and explored in B-E. (B) GO enrichment of significant DEGs between wild type and *Hhip*⁻/⁻ in the coronal and lambdoid sutures. Points are colored according to the alluvial flows in A and sized based on the number of DEGs annotated with the term. (C) GO enrichment of significant DEGs between wild type and *Hhip*⁻/⁻ unique to the coronal or lambdoid sutures. Points are colored according to the alluvial flows in A and sized based on the number of DEGs annotated with the term. (D) Scatterplot comparing the fold-change in gene expression between *Hhip*⁻/⁻ and wild type for all coronal (*x*-axis) and lambdoid suture DEGs (*y*-axis). Significant DEGS with a >1.75-fold change between *Hhip*⁻/⁻ and wild type in both sutures are highlighted and colored according to the alluvial flows in A. (E) As in D but highlighting HH pathway genes and regulators of *Ihh* expression. (F) Volcano plot of the DEGs between wild-type coronal and lambdoid sutures at E18.5.

## Comparison of transcriptome changes between *Hhip*⁻/⁻ coronal and lambdoid sutures

To identify changes in *Hhip* mutants that were common to both sutures, or unique to each suture, we performed pairwise comparisons of DEGs between *Hhip*⁻/⁻ coronal and lambdoid sutures (compared to wild type) (Fig. 4A,D; Table S4). Genes upregulated in both *Hhip*⁻/⁻ sutures (132 genes) were enriched for GO BP terms such as 'regulation of response to stimulus', 'regulation of multicellular organismal process', 'cell communication' and 'signaling' (Fig. 4B; Table S4). Genes downregulated in both *Hhip*⁻/⁻ sutures (77 genes) were enriched for GO BP terms such as 'tissue development' and 'anatomical structure morphogenesis' (Fig. 4B; Table S4). Of the common up- or downregulated genes many were associated with skeletal development in mice. Notably, common downregulated genes included *Sgms2* and *Smpd3*, which are expressed in osteoblasts and act in opposition in the interconversion of sphingomyelin and ceramide, a process that promotes bone mineralization (Qi et al., 2021). The retinoic acid-degrading enzyme *Cyp26b1*, loss of which can cause CS in humans and zebrafish (Laue et al., 2011), and reduced calvarial ossification in mice (Maclean et al., 2009), was downregulated in both *Hhip*⁻/⁻ sutures. Consistent with histological data, proliferation markers were not altered in E18.5 *Hhip*⁻/⁻ coronal (Holmes et al., 2021) or lambdoid (Fig. 2) sutures.

Genes that were upregulated in the *Hhip*⁻/⁻ coronal suture and unchanged in the lambdoid (505 genes; Fig. 4A; Table S4) were enriched for GO BP terms such as 'cell differentiation', 'cellular developmental process', and 'actin filament-based process' (Fig. 4C; Table S4). Genes that were downregulated in the *Hhip*⁻/⁻ coronal suture and unchanged in the lambdoid (353 genes; Fig. 4A; Table S4) were enriched for the GO BP term 'cytoplasmic translation' and related terms, driven by decreased expression of ribosomal large and small protein genes, and WNT signaling (Fig. 4C; Table S4). Genes that were upregulated only in the lambdoid suture and unchanged in the coronal (633 genes; Fig. 4A; Table S4) were enriched for GO BP terms such as 'immune system process', 'locomotion' and 'response to stimulus' (Fig. 4C; Table S4). Genes that were downregulated in the lambdoid suture and unchanged in the coronal (390 genes; Fig. 4A; Table S4) were enriched for GO BP terms such as 'generation of neurons' and similar terms (Fig. 4C; Table S4). Genes that were downregulated in the coronal suture but upregulated in the lambdoid (71 genes; Fig. 4A; Table S4) were enriched for GO BP terms such as 'bone resorption' and related terms, driven by changes in expression of osteoclast-specific genes (Fig. 4B; Table S4). No GO BP terms were associated with genes that were upregulated in the coronal suture but downregulated in the lambdoid suture (18 genes; Fig. 4A; Table S4).

## Transcriptome differences between wild-type coronal and lambdoid sutures

As the wild-type coronal and lambdoid sutures differ structurally and developmentally, we compared their transcriptomes. We identified 3142 DEGs, of which 1647 were more highly expressed in the coronal suture and 1495 were more highly expressed in the lambdoid suture (FDR<0.05) (Fig. 4F; Table S3). By GO analysis, coronal DEGs were enriched for many BP terms, including those related to cell migration, 'blood vessel development' and 'osteoclast differentiation' (Table S3). The most enriched KEGG terms were related to 'osteoclast differentiation' (Table S3). These GO and KEGG enrichments were similar to those in the upregulated genes of the *Hhip*⁻/⁻ lambdoid suture. Lambdoid DEGs were most enriched for GO BP terms related to 'synapse organization' and mast cells (Table S3). Enriched KEGG terms included 'neuroactive ligand-receptor interaction' and 'renin-angiotensin system', the latter term driven by increased expression of mast cell markers (Table S3).

Of the coronal DEGs, eight of the top 15 were osteoclast markers (Table S3). Other coronal DEGs associated with craniofacial or bone defects include *Pitx2*, *Sim2* and *Calcr*. The lambdoid DEGs whose mutation in mouse results in craniofacial phenotypes included *Hoxb2*, *Mab21l2*, *Grm3* and *Pax1*. Overall, these results show that the coronal and lambdoid sutures differ in their basal expression of genes that influence bone development and in resident cell populations, with osteoclasts enriched in the coronal suture and mast cells enriched in the lambdoid suture.

We also assessed differential expression of HH pathway genes and genes regulating *Ihh* expression between the wild-type coronal and lambdoid sutures (Table 1; Table S3). *Ihh*, *Gli1*, *Ptch1* and *Ptch2* were not DEGs. Expression of *Gli3* has been reported to be higher in frontal and lambdoid sutures (Rice et al., 2010), but was expressed at similar levels in the coronal and lambdoid sutures in our data. Expression of *Hhip*, *Pthlh* and *Scube1* was higher in the coronal suture, with *Pthlh* being the strongest DEG assessed (log2 FC=4.70 compared to the lambdoid). Expression of *Cdon*, *Disp1*,

*Gas1*, *Ptchd4*, *Smo*, *Scube2* and *Scube3* was higher in the lambdoid suture to varying degrees (Table 1 and Table S3). These differences suggest a potential for suture-specific variation in HH signaling strength and response between the two sutures.

## *Pthlh* expression depends on IHH expression and prevents fusion of the *Hhip*⁻/⁻ coronal suture

The decrease in *Ihh* expression in *Hhip*⁻/⁻ OFs was a surprising result, given the clear increase in HH signaling in both sutures. *Ihh* expression is highest in the early osteoblasts of the OFs and downregulated in mature osteoblasts. Its decrease in *Hhip*⁻/⁻ OFs suggests that *Ihh* expression specifically, or the state of osteogenic differentiation permitting *Ihh* expression, is diminished. We considered *Pthlh*, the expression of which increased in both *Hhip*⁻/⁻ sutures, a candidate for this activity. In addition, there is a much higher level of *Pthlh* expression in the coronal suture, which may further account for the less severe phenotype in the *Hhip*⁻/⁻ coronal suture compared to the lambdoid. In reviewing our single-cell RNA-seq data of the wild-type coronal suture (Holmes et al., 2021), we found that *Pthlh* expression was enriched in the same SM population expressing *Hhip*. *Pthlh* expression in the coronal SM also has been reported previously (Farmer et al., 2021). In the growth plate of long bones, PTHLH secreted from the resting chondrocytes at the end of the growth plate delays the differentiation of PTH1R-expressing, proliferating chondrocytes to prehypertrophic and hypertrophic chondrocytes, which express *Ihh* (Vortkamp et al., 1996). *Pthlh* expression in turn depends on IHH (St-Jacques et al., 1999). This signaling loop regulates the rate of growth plate lengthening. Loss of *Pthlh* (Vortkamp et al., 1996) or *Pth1r* (Lanske et al., 1996) expression does not clearly affect *Ihh* expression levels, but results in a shift of the *Ihh*-expressing chondrocytes towards the growth plate. Treatment of cultured growth plates with PTHLH can eliminate *Ihh* expression (Vortkamp et al., 1996).

We therefore postulated a regulatory relationship between *Ihh* and *Pthlh* in the sutures. In both the wild-type and *Hhip*⁻/⁻ coronal

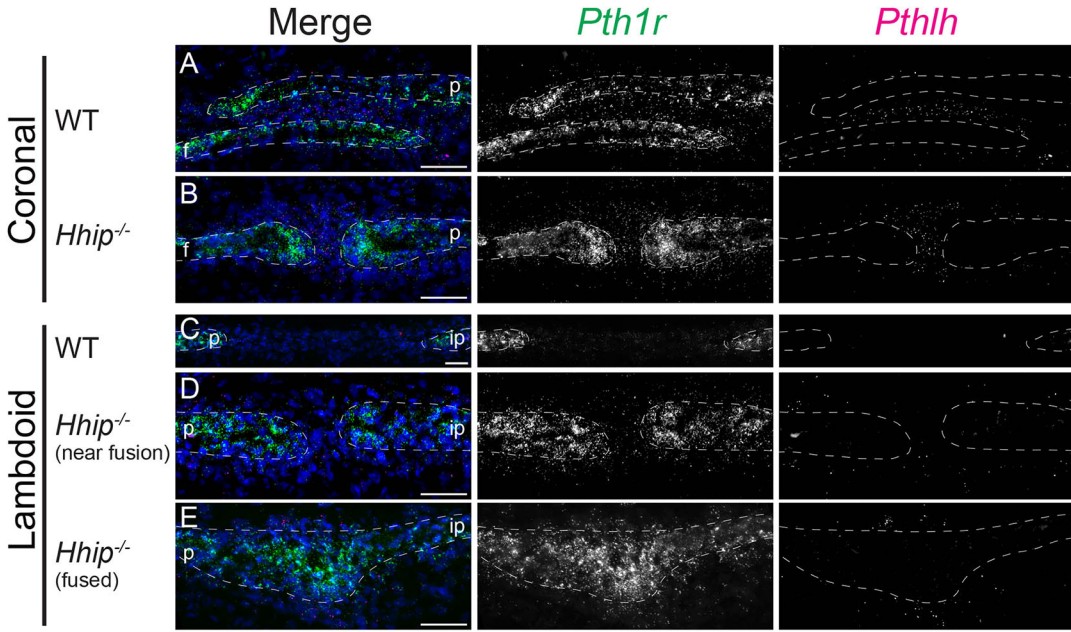

**Fig. 5. *Pthlh* expression in the E18.5 coronal SM shifts relative to the osteogenic fronts in the *Hhip*⁻/⁻ coronal suture.** smFISH for co-expression of *Pth1r* (green) and *Pthlh* (magenta), counterstained with DAPI (blue) in (A) wild-type coronal, (B) *Hhip*⁻/⁻ coronal, (C) wild-type lambdoid, (D) *Hhip*⁻/⁻ lambdoid near fusion and (E) *Hhip*⁻/⁻ lambdoid fused region. Gene colors in merged images are indicated by text color above greyscale images. *n*=3 wild type and 3 *Hhip*⁻/⁻. White dashed lines indicate frontal (f), interparietal (ip) and parietal (p) bones. Sections are in the transverse plane. Scale bars: 50 µm.

suture at E18.5, *Pth1r* expression was enriched in the frontal and parietal bones, with lower expression in the SM (Fig. 5A,B). In the wild-type suture, *Pthlh* expression was enriched in the SM in the region of bone overlap and decreased proximal to the OFs (Fig. 5A). This expression was similar to but more restricted to SM than that of *Hhip* (Fig. 3D) (Holmes et al., 2021). In the *Hhip*$^{-/-}$ suture, *Pthlh* expression remained in the SM, but was now proximal to and surrounding the closely opposed frontal and parietal OFs (Fig. 5B). In the wild-type and unfused *Hhip*$^{-/-}$ lambdoid suture, *Pth1r* expression was enriched in the parietal and interparietal bones, with lower expression in the SM (Fig. 5C,D). In contrast to the coronal suture, *Pthlh* expression was undetectable in the wild-type lambdoid suture (Fig. 5C) and low in the *Hhip*$^{-/-}$ lambdoid suture (Fig. 5D), consistent with their relative RNA-seq expression values (Table 1 and Table S3). In the fused lambdoid suture, *Pth1r* expression was continuous across the fusion point, and *Pthlh* expression was detectable in adjacent mesenchyme (Fig. 5E).

To determine whether these changes correlated with a shift in the relative location of *Ihh* expression, we performed smFISH for *Ihh*, *Sp7* (first expressed in committed preosteoblasts) and *Pthlh* (Fig. 6). In the wild-type E18.5 coronal and lambdoid suture OFs, *Ihh* expression began in the OFs immediately behind the start of the *Sp7* expression domain (Fig. 6A,C). In the *Hhip*$^{-/-}$ coronal and lambdoid suture OFs, decreased *Ihh* expression was less consistently close to the start of *Sp7* expression, but was not always displaced (Fig. 6B,D). Where the *Hhip*$^{-/-}$ lambdoid suture fused, both *Sp7* and *Ihh* were expressed throughout the bone (Fig. 6E). These results suggest that the increased and shifted (in the coronal suture) *Pthlh* expression did not displace the *Ihh*-expressing population relative to the *Sp7*-expressing population, but specifically downregulated *Ihh* expression and/or altered the differentiation state of osteoblasts permissive for *Ihh* expression in the OF.

In the growth plate, *Pthlh* expression depends on the expression of *Ihh* (St-Jacques et al., 1999). IHH also induces *Pthlh* expression in mature postnatal osteoblasts (Mak et al., 2008). In *Ihh*$^{-/-}$

embryos, coronal and lambdoid sutures appear wider in whole calvaria stained for mineralized bone (Abzhanov et al., 2007; Klopocki et al., 2011; Veistinen et al., 2017). We confirmed histologically that E18.5 *Ihh*$^{-/-}$ coronal and lambdoid suture development was impaired (Fig. 7 and Fig. S3). Mutant coronal sutures typically lacked the overlap between frontal and parietal bones (Fig. 7A,B and Fig. S3A,B,D). Mutant OFs in both sutures were also thinner than wild type, and ALP activity was weaker throughout the calvarial bones (Fig. S3), in agreement with a previous report of decreased *Alp* expression (Lenton et al., 2011). Expression of the HH transcriptional targets *Gli1*, *Hhip* and *Ptch1* was absent in *Ihh*$^{-/-}$ coronal sutures (Fig. 7C-F). *Pthlh* expression also was not detected compared to wild type (Fig. 7E,F), indicating that its expression depends on *Ihh* transcription.

Finally, to test whether *Pthlh* expression prevents fusion of the *Hhip*$^{-/-}$ coronal suture, we generated compound heterozygous *Hhip*$^{+/-}$;*Pthlh*$^{+/-}$ mice and crossed these to obtain E18.5 litters containing double-knockout *Hhip*$^{-/-}$;*Pthlh*$^{-/-}$ embryos. *Pthlh*$^{-/-}$ coronal sutures were grossly normal (Fig. 8A,B), with levels of *Ihh*, *Gli1*, *Hhip* and *Ptch1* expression comparable to wild type (Fig. 8F, G,K,L). In contrast to the *Hhip*$^{-/-}$ coronal suture, *Hhip*$^{-/-}$;*Pthlh*$^{-/-}$ coronal sutures showed varied degrees of bilateral (6/6) fusion (Fig. 8C-E). Fusion, determined by the presence of continuous osteoid between frontal and parietal sutures in histological sections, spanned ~49±20% (mean±s.d.; *n*=12) of the length of individual coronal sutures. Deletion of both genes was required for fusion by E18.5, as *Hhip*$^{+/-}$;*Pthlh*$^{-/-}$ and *Hhip*$^{-/-}$;*Pthlh*$^{+/-}$ sutures did not fuse (Fig. S4). *Hhip*$^{-/-}$;*Pthlh*$^{-/-}$ lambdoid sutures also showed significantly more fusion than *Hhip*$^{-/-}$ lambdoid sutures (49±14%, *n*=12, compared to 33±12%, *n*=8; unpaired *t*-test, *P*=0.018). The domain and intensity of *Ihh* OF expression in regions of incompletely fused *Hhip*$^{-/-}$;*Pthlh*$^{-/-}$ coronal sutures appeared qualitatively increased compared to *Hhip*$^{-/-}$, although clearly not to wild-type levels (Fig. 8F,H,I,K,M,N). *Gli1* and *Ptch1* expression were comparable or higher (Fig. 8H,I,M,N). In fused regions, *Ihh*

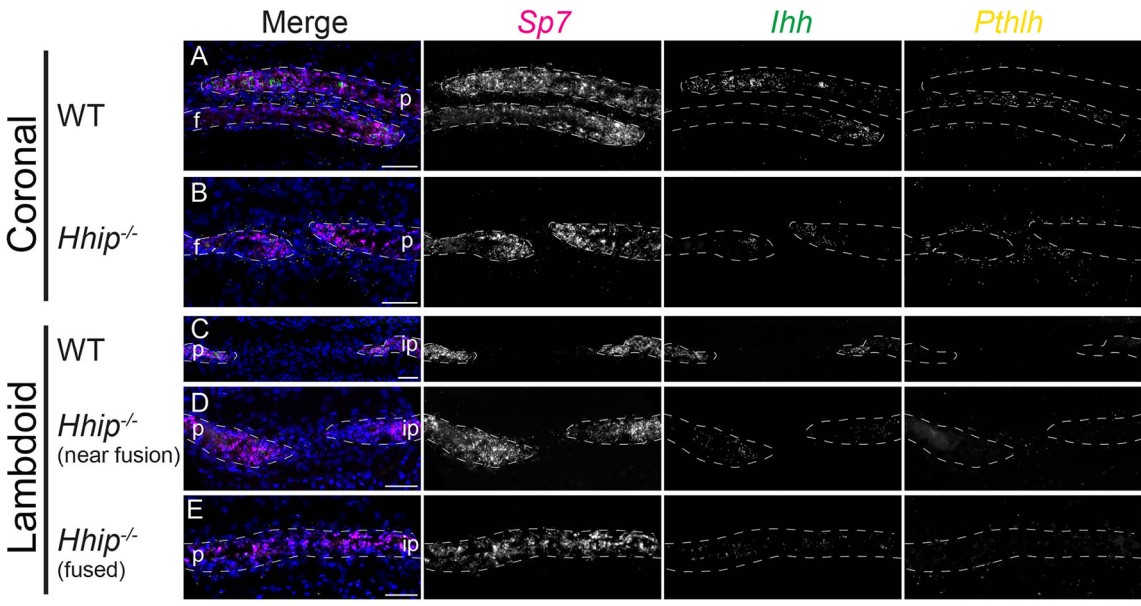

**Fig. 6. Relative expression domains of *Sp7* and *Ihh* are similar in the E18.5 wild-type and *Hhip*$^{-/-}$ coronal and lambdoid sutures.** (A-E) smFISH for co-expression of Sp7 (magenta), *Ihh* (green) and *Pthlh* (yellow), counterstained with DAPI (blue) in (A) wild-type coronal, (B) *Hhip*$^{-/-}$ coronal, (C) wild-type lambdoid, (D) *Hhip*$^{-/-}$ lambdoid near fusion and (E) *Hhip*$^{-/-}$ lambdoid fused region. Gene colors in merged images are indicated by text color above greyscale images. *n*=3 wild type and 3 *Hhip*$^{-/-}$. White dashed lines indicate frontal (f), interparietal (ip) and parietal (p) bones. Sections are in the transverse plane. Scale bars: 50 μm.

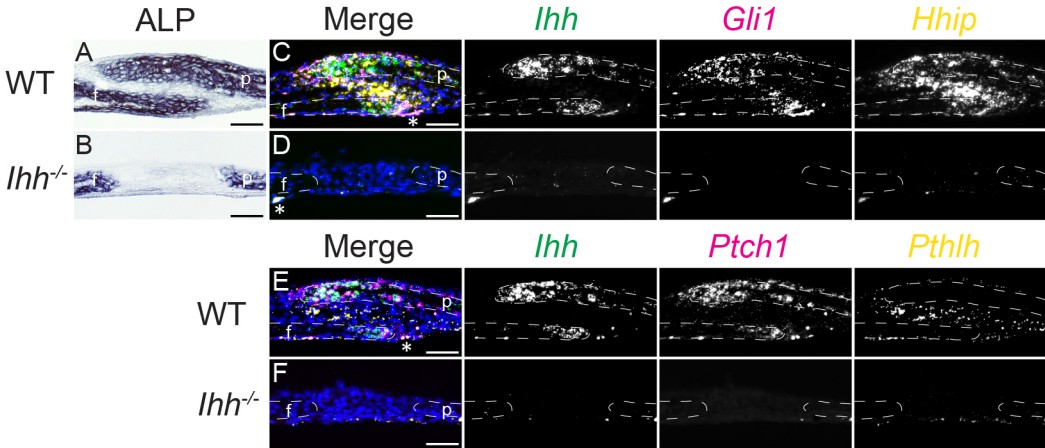

**Fig. 7. *Pthlh* expression is dependent on *Ihh* in the E18.5 coronal suture.** (A,B) ALP activity (blue) in wild-type (A) and *Ihh*$^{-/-}$ (B) coronal sutures. (C-F) smFISH for co-expression of (C,D) *Ihh* (green), *Gli1* (magenta) and *Hhip* (yellow), and (E,F) *Ihh* (green), *Ptch1* (magenta) and *Pthlh* (yellow), counterstained with DAPI (blue). Gene colors in merged images are indicated by text color above greyscale images. *n*=5 wild type and 7 *Ihh*$^{-/-}$. White dashed lines indicate frontal (f) and parietal (p) bones. Asterisks in merged images indicate non-specific fluorescence. Sections are in the sagittal plane. Scale bars: 50 μm.

expression remained across the fusion point (Fig. 8J,O), and *Gli1* and *Ptch1* were expressed in the newly fused bone and adjacent mesenchyme at higher levels than in the *Hhip*$^{-/-}$ coronal suture (Fig. 8H,J,M,O). These results support a role for *Pthlh* in the direct or indirect repression of *Ihh* transcription at the OFs, and in the negative regulation of HH signaling in combination with *Hhip* to maintain a patent coronal suture (Fig. 9).

## DISCUSSION

Hedgehog signaling mediated by IHH plays a significant role in suture development (Ishii et al., 2015). Dysregulated activation of HH signaling can cause CS, but how HH signaling is constrained during suture development is not fully understood. We previously identified loss of SM but not fusion of coronal sutures in E18.5 *Hhip*$^{-/-}$ embryos (Holmes et al., 2021). Here, we demonstrate that the *Hhip*$^{-/-}$ lambdoid suture undergoes fusion by E18.5. An RNA-seq analysis of *Hhip*$^{-/-}$ coronal and lambdoid sutures identified common and distinct transcriptional changes within each suture, including upregulation of HH transcriptional targets in both sutures, despite decreased *Ihh* expression. An explanation for this may be that any loss of IHH protein resulting from *Ihh* downregulation is compensated for by the failure to neutralize IHH by HHIP. Intriguingly, *Pthlh* expression is upregulated in both *Hhip*$^{-/-}$ sutures, and its shift in location to coronal SM adjacent to *Ihh* expression in OFs suggests an inhibitory effect on *Ihh* expression reminiscent of that occurring in cartilage growth plates. Its much higher expression in the coronal compared to the lambdoid suture suggests it could have a greater biological role in the coronal suture, preventing outright fusion in the absence of *Hhip*. As in the growth plate, we demonstrate that sutural *Pthlh* expression is dependent on *Ihh* expression. Finally, we demonstrate that the combined loss of *Hhip* and *Pthlh* results in coronal suture fusion by E18.5. Loss of *Pthlh* also augments the extent of *Hhip*$^{-/-}$ lambdoid suture fusion, suggesting a similar but less crucial role in the lambdoid suture. We therefore identify a previously unreported *Ihh*/*Pthlh* regulatory interaction of particular significance to the coronal suture.

Increased bone growth was evident in the skull of *Hhip*$^{-/-}$ embryos, the size of which was larger overall, with the frontal and parietal bones having a greater volume and thickness. Additionally, ectopic bone formation occurred at the lower posterior margin of the parietal bone, and in the lambdoid suture, which underwent fusion

by E18.5. Lambdoid suture fusion is uncommon in humans (Heuzé et al., 2014). However, in two mouse lines with mutations of *Ptch1* (Feng et al., 2013) or *Gli3* (Rice et al., 2010) that activate HH signaling and have reported CS, the lambdoid suture is principally affected. Together with the *Hhip*$^{-/-}$ mouse, this suggests that the murine lambdoid suture may be more susceptible to fusion when HH signaling is increased.

The difference in *Hhip*$^{-/-}$ phenotype between lambdoid and coronal sutures raised questions about the regulatory differences between these two sutures, in particular about HH signaling. Wild-type coronal and lambdoid sutures had distinct gene expression profiles in addition to their different physical arrangement of bones, which together may influence the response of each suture to genetic perturbation. In particular, they differed in the expression levels of regulatory components of the HH pathway, which could result in complex differences in response to changes in HH signaling, such as in the *Hhip*$^{-/-}$ embryos. Of these, we focused on *Pthlh*, which was the most differentially expressed HH-related gene between the two sutures, and was notable for its expression throughout the coronal SM.

The coronal and lambdoid sutures also differ in their relative contributions of neural crest and mesoderm lineages. In the coronal suture the frontal bone is derived from neural crest, while the SM and parietal bone are derived from mesoderm; in the lambdoid suture, the parietal bone and SM are derived from mesoderm, while the interparietal bone is derived mainly from mesoderm but with a central portion derived from neural crest (Jiang et al., 2002; Yoshida et al., 2008). The lateral regions of the lambdoid suture fused at E18.5 in our study are derived from mesoderm. The potential contribution of cell lineage on the differing sensitivity to loss of *Hhip* of the two sutures is unclear. Osteoblasts from frontal bones have been characterized as having greater proliferative, osteogenic and regenerative capacities than parietal osteoblasts, correlating with differences in FGF, BMP, WNT and TGFß signaling (Doro et al., 2019; Li et al., 2010; Menon et al., 2021). A few studies have defined global transcriptional differences between frontal and parietal bones, but it is unclear whether these are lineage-specific or bone-specific differences (Chen et al., 2019; Homayounfar et al., 2015). The two genes that are the focus of our study, *Hhip* and *Pthlh*, are expressed in mesoderm-derived tissues in both sutures, while being more highly expressed in the coronal suture (Table 1), and so differences in lineage between the sutures would not appear to be a

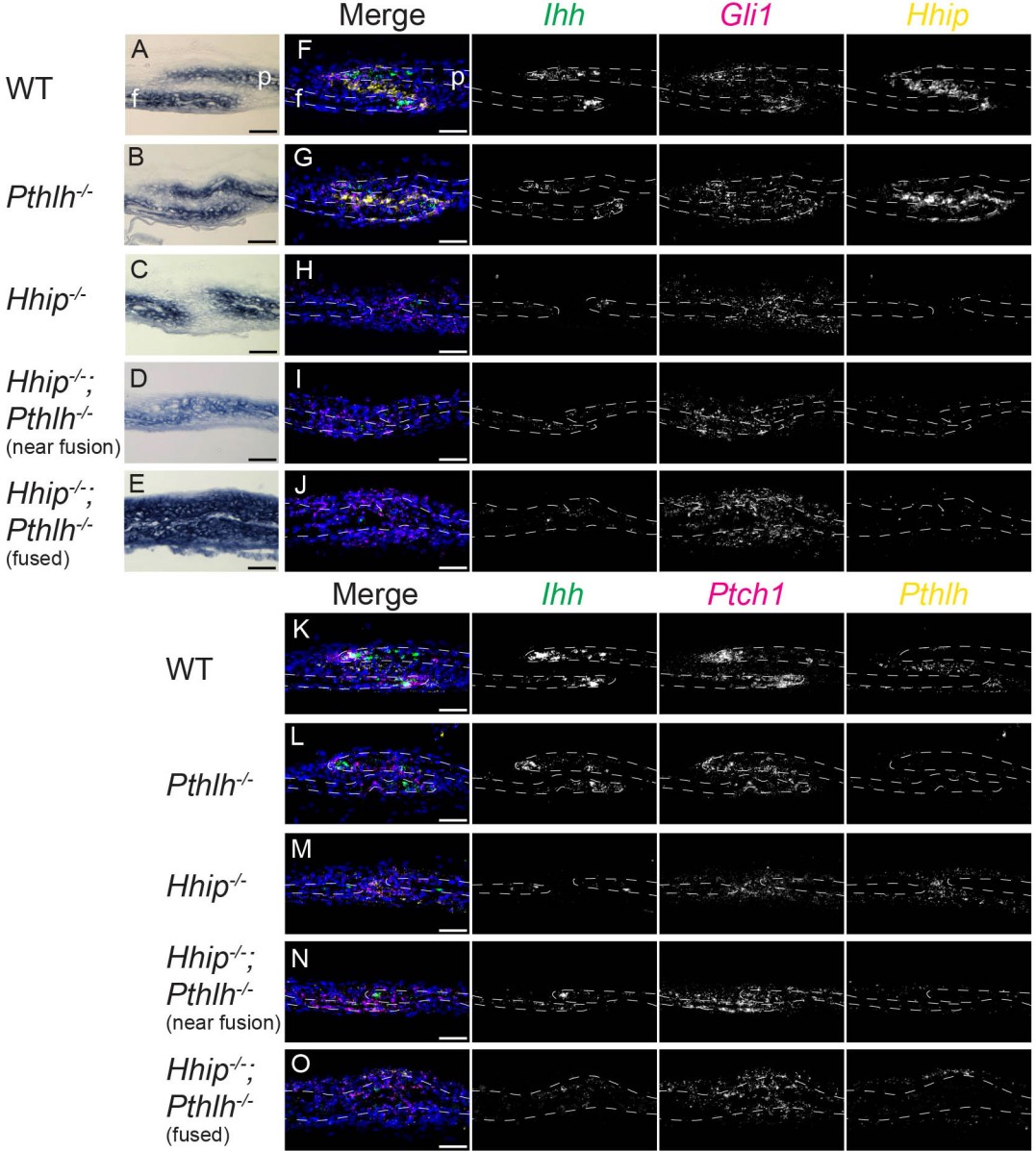

**Fig. 8. *Pthlh* prevents coronal suture fusion in the absence of *Hhip*.** (A-E) ALP activity (blue) and (F-O) smFISH for co-expression of the indicated HH pathway genes in wild-type (A,F,K; *n*=2), *Pthlh$^{-/-}$* (B,G,L; *n*=3), *Hhip$^{-/-}$* (C,H,M; *n*=4), *Hhip$^{-/-}$;Pthlh$^{-/-}$* near fusion (D,I,N) and *Hhip$^{-/-}$;Pthlh$^{-/-}$* fused (E,J,O; *n*=4) E18.5 coronal sutures. Gene colors in merged images are indicated by text color above greyscale images. White dashed lines indicate frontal (f) and parietal (p) bones. Sections are in the sagittal plane. Scale bars: 50 μm.

factor in their differing response to loss of *Hhip*. Gene expression differences may also reflect the differing structure of the two sutures. However, the lack of fusion in neural crest-derived *Hhip$^{-/-}$* facial sutures with either overlapping or end-to-end structures (Fig. S2) suggests that neither lineage nor structure alone are determinants of susceptibility to fusion in the absence of *Hhip*.

Another distinction between wild-type coronal and lambdoid sutures was the enrichment of osteoclasts in the coronal suture, evident by their marker gene expression. Interestingly, the *Hhip$^{-/-}$* lambdoid suture also was enriched for osteoclasts. Given the coupling of osteogenesis and osteoclastogenesis (Zaidi et al., 2023), we suggest that this is a consequence of the advanced osteogenesis. This plausibly can be linked to increased HH signaling and *Pthlh* expression. In mature osteoblasts, IHH induces *Pthlh* expression, and PTHLH upregulates *Tnfsf11* (*Rankl*) expression to induce osteoclast differentiation (Mak et al., 2008). *Rankl* and *Csf1*, which

also encodes an inducer of osteoclast differentiation, were upregulated in the *Hhip$^{-/-}$* lambdoid suture.

How PTHLH regulates HH signaling in the suture is currently unclear. PTH/PTHLH signaling can inhibit the HH pathway directly or indirectly. PTHLH activates PTH1R, a G protein-coupled receptor. One result is the activation of the G protein subunit GαS and the cAMP/PKA pathway, which regulates the activity of GLI2 and GLI3 to inhibit the HH pathway (He et al., 2014; Regard et al., 2013). In the calvaria, widespread deletion of GαS activity increased expression of HH target genes without affecting *Ihh* expression, while activation of GαS decreased HH target gene expression, although the effect on *Ihh* expression was not reported (Xu et al., 2018). However, in both *Hhip$^{-/-}$* sutures, HH signaling and target expression is increased, despite increased *Pthlh* expression, suggesting that PTHLH is not directly inhibiting GLI activity. *Pth1r* is expressed at low levels in the SM and at

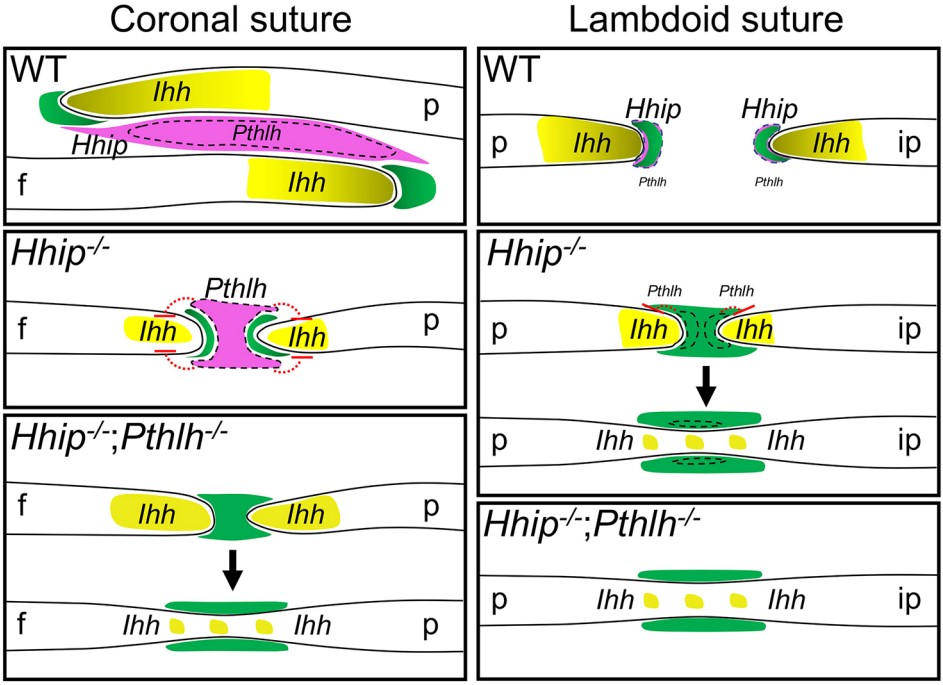

**Fig. 9. Model of coronal and lambdoid suture regulation by *Hhip* and *Pthlh*.** In the wild-type coronal suture (top left), *Ihh* expressed in the osteogenic fronts (OFs) (yellow) induces HH-mediated osteogenic activity in suture mesenchyme (SM) adjacent to the OFs (green). In the remaining SM, osteogenic activity is limited (magenta) by *Ihh*-mediated induction of *Hhip* expression (entire magenta region). *Pthlh* expression is also induced (dashed outline), but its function in the wild-type suture is unclear. In the wild-type lambdoid suture (top right), osteogenic activity induced by *Ihh* in adjacent SM is limited by co-expression of *Hhip*. *Pthlh* expression, here proposed to be in a similar domain to *Hhip*, is much lower compared to the coronal suture. In the $Hhip^{-/-}$ coronal suture (left, middle), increased HH-mediated osteogenesis prevents overlap of the frontal and parietal bones, but *Pthlh* expression is increased in closer proximity to the OF cells expressing *Ihh* and reduces *Ihh* expression, limiting osteogenesis in the remaining SM. In the $Hhip^{-/-}$ lambdoid suture (right, middle), HH-mediated osteogenesis extends throughout the SM, leading to fusion, despite increased *Pthlh* expression and reduced *Ihh* expression. In the $Hhip^{-/-};Pthlh^{-/-}$ coronal suture (bottom left), the absence of *Hhip* and *Pthlh* allows osteogenic HH signaling throughout the SM, resulting in fusion. In the $Hhip^{-/-};Pthlh^{-/-}$ lambdoid suture (bottom right), fusion is more extensive than in the $Hhip^{-/-}$ suture, suggesting a similar but much less significant role for *Pthlh* compared to its role in the coronal suture. f, frontal bones; ip, interparietal bones; p, parietal bones.

much higher levels in osteoblasts, so the expected site of *Pthlh* signaling would be osteoblasts of the OF and more mature bone. Furthermore, HH target expression in $Hhip^{-/-}$ sutures is increased largely in the SM, outside of the region of enriched *Pthr1r* expression, so the apparent domains of PTHLH and HH signaling do not overlap greatly. Together, this suggests that PTHLH acts on osteoblasts expressing *Ihh*, rather than inhibiting signaling in the SM. In the growth plate, PTHLH inhibits *Ihh* expression indirectly by delaying the differentiation of proliferating chondrocytes to prehypertrophic chondrocytes, which express *Ihh* (Vortkamp et al., 1996). In $Hhip^{-/-}$ sutures, we noted some displacement of *Ihh* expression from the border of *Sp7*-expressing preosteoblasts, but this was inconsistent. Alternatively, exposure of murine growth plates to PTHLH in culture eliminated *Ihh* expression (Vortkamp et al., 1996), so direct inhibition of *Ihh* expression by PTHLH is possible. This mechanism is supported by the finding that treatment of chondrocytes with PTH or PTHLH downregulates *Ihh* expression (Yoshida et al., 2001). In $Hhip^{-/-}$ sutures, increased PTHLH may therefore be downregulating *Ihh* expression directly or more broadly inhibiting aspects of early osteoblast differentiation that reduce *Ihh* expression.

Increased HH signaling resulting from loss of *Hhip* expression could promote osteogenesis in the suture through additional pro-osteogenic signaling pathways that regulate suture development. IHH promotes the expression and activity of RUNX2 at the OFs (Veistinen et al., 2017). A study of the postnatal posterior frontal suture showed that RUNX2 positively regulates the expression of

effector genes of the HH, FGF, WNT and PTH pathways (Qin et al., 2019). In turn, these pathways can positively regulate RUNX2 (Komori, 2024; Teplyuk et al., 2009). In particular, aberrant FGF signaling caused by activating mutations of FGF receptors (FGFRs), which are expressed at the OFs, is a major cause of syndromic CS (Heuzé et al., 2014; Wilkie et al., 2017). However, our RNA-seq analyses of $Hhip^{-/-}$ sutures did not show significant changes in FGF ligand or receptor expression, apart from a decrease in *Fgf10* expression in both sutures (Table S4).

HHIP and PTHLH have clinical relevance. Variants within regulatory sequences of HHIP expression are associated with chronic obstructive pulmonary disease (COPD) (Lahmar et al., 2022; Wilk et al., 2009). GWAS studies have identified single nucleotide polymorphisms (SNPs) near the HHIP locus associated with hip shape (Baird et al., 2019; Styrkarsdottir et al., 2019) and height (Liu et al., 2010). Loss-of-function mutations of PTHLH cause brachydactyly type E (Klopocki et al., 2010; Scheffer-Rath et al., 2023), which may include craniofacial dysmorphism (Jamsheer et al., 2016), while PTHLH duplications cause chondrodysplasia (Echaubard et al., 2022; Flöttmann et al., 2016; Gray et al., 2014). PTH1R mutations cause skeletal dysplasias, including Jansen's metaphyseal chondrodysplasia, Eiken syndrome and Blomstrand chondrodysplasia (Cheloha et al., 2015). In combination, genetic variants affecting the expression or function of both genes or their signaling pathways could affect suture, craniofacial or skeletal development more strongly than individual variants alone.

## MATERIALS AND METHODS

### Mice

Mouse procedures complied with guidelines of the Institutional Animal Care and Use Committee (IACUC) of the Icahn School of Medicine at Mount Sinai. Timed matings of heterozygous $Hhip^{tm1Amc}$/J mice (Chuang et al., 2003) (The Jackson Laboratory, strain 006241; on a C57BL/6, Swiss-Webster, 129 background; homozygotes referred to as $Hhip^{-/-}$), heterozygous $Pthrp^{mCherry}$ mice (Mizuhashi et al., 2018) (The Jackson Laboratory, strain 032872; on a C57BL/6 background; homozygotes referred to by their official gene name as $Pthlh^{-/-}$), double heterozygous $Hhip^{+/-};Pthlh^{+/-}$ mice or heterozygous $Ihh^{tm1Amc}$ mice (St-Jacques et al., 1999) (The Jackson Laboratory, strain 004290; on a mixed C57BL/6, 129 background; homozygotes referred to as $Ihh^{-/-}$) were performed to obtain embryos at the required ages. Genotyping was performed by polymerase chain reaction of tail DNA. $Hhip$, $Pthlh$ and $Ihh$ genotypes were identified as described by The Jackson Laboratory. Sex genotypes were identified as described previously (Bean et al., 2001; Wang et al., 2010).

### Immunohistochemistry and cytochemistry

Immunohistochemistry and EdU staining were performed on 10 µm sections from either fresh frozen or 4% PFA-fixed cryoembedded heads prepared as previously described (Holmes et al., 2018). Antibody staining for RUNX2 (1:200; rabbit anti-RUNX2, Sigma-Aldrich, HPA022040), SP7 (1:500; rabbit anti-SP7/Osterix, Abcam, ab22552), and Cleaved Caspase-3 (1:400, rabbit-anti-Cleaved Caspase-3, Cell Signaling Technology, 9661S) was performed after ALP staining using standard procedures. Primary antibodies were detected with donkey anti-rabbit IgG Alexa Fluor 488 (1:400; Thermo Fisher Scientific, A-21206). For EdU staining, pregnant mice were injected with EdU (250 µg/10 g body weight) 2 h before being euthanized. EdU staining was performed with the Click-iT Plus EdU Alexa Fluor 488 Imaging Kit (Thermo Fisher Scientific, C10637), as described by the manufacturer. Sections were counterstained with DAPI. EdU-positive nuclei and total nuclei were counted within the individual OFs and SM of five non-consecutive sections per lambdoid suture and averaged. In $Hhip^{-/-}$ sutures, only unfused regions adjacent to fused regions were assessed. In OFs, nuclei were counted within the ALP-positive domain between the SM and the start of the osteoid, or within a 50 µm distance if the osteoid was not apparent. SM was defined as ALP-negative cells between the ends of the ALP-positive OFs. Cell counts were performed in Adobe Photoshop. Images were taken on a Nikon Eclipse E600 microscope equipped with a Nikon DS-Ri2 digital camera and NIS Elements (F4.30.01) software.

### Histochemical staining

For alkaline phosphatase (ALP) staining to image fluorescent red signal, sections were stained using 0.4 mg/ml Fast Red TR salt (Sigma-Aldrich, F6760), 0.1 M tris-maleate buffer (pH 9.2) (Sigma-Aldrich, T3128) and 0.02% naphthol AS-MX (Sigma-Aldrich, 855) for a few seconds and washed in 1×PBS three times for 5 min each before being used in further assays (Holmes et al., 2020). Nuclei were counterstained with DAPI. For ALP chromogenic staining, sections were stained using NBT/BCIP (Sigma-Aldrich/Roche, 11681451001) as described by the manufacturer. For LacZ staining, fresh frozen sections were fixed in 0.2% glutaraldehyde (Sigma-Aldrich, G5882) for 10 min. Slides were then incubated overnight in the dark at 37°C in 1 mg/ml of X-galactosidase in staining buffer (200 mM $K_3Fe$; 200 mM $K_4Fe$; 1 M $MgCl_2$; 10% NP40; 1× PBS). Sections were counterstained with nuclear Fast Red (Vector Laboratories, H-3403). ALP red fluorescence and LacZ images were taken on a Nikon Eclipse E600 microscope equipped with a Nikon DS-Ri2 digital camera and NIS Elements software (F4.30.01). ALP blue chromogenic images were taken on either a Nikon Eclipse 80i microscope equipped with a Nikon DS-Fi3 digital camera and NIS Elements software (F5.20.01) (Fig. S3) or an AxioImager Z2 M equipped with a monochrome Axiocam 503 camera and Zen 2 Blue Edition software (version 2.0) (Figs 7 and 8).

### Histological determination of the extent of suture fusion

For $Ihh$ knockout and $Hhip/Pthlh$ double knockout studies, ALP staining and smFISH were performed on calvaria stacked and embedded in a single OCT block. Fixed calvariae of the required genotypes were bisected along the midline, and left and right hemi-calvariae were trimmed to rectangular coupons that contained the coronal and lambdoid sutures from the midline to their lateral limits. Left and right halves were then embedded in separate blocks, with the hemi-calvariae stacked in a recorded order, with up to five hemi-calvariae in a stack. The midline edge of each hemi-calvaria was aligned along a wall of the plastic mold to maintain positional registration between each hemi-calvaria in the stack and to facilitate identification of the tissue for sectioning at 10 µm in the sagittal plane.

Hemi-calvariae stacks were sectioned completely from the midline to the lateral edge. Alternate sections were collected in sets of 10 slides, with 10 sections per slide; 3 sets of 10 slides were typically sufficient to span the entire hemi-calvariae. One slide from each set was stained for ALP to assess suture patency and to determine which slides were optimal for smFISH. To determine the extent of coronal or lambdoid suture fusion, ALP-stained sections from one slide in each set were assessed. Sutures were scored as unfused if osteoid was discontinuous, or as fused if osteoid was continuous, between the frontal and parietal bones (coronal suture) or the parietal and interparietal bones (lambdoid suture). The degree of fusion was expressed as a percentage of fused sutures to total sutures (fused plus unfused) scored.

### Microcomputed tomography (microCT)

E18.5 wild-type and $Hhip^{-/-}$ embryos were fixed in 4% PFA for 48 h, equilibrated in PBS and stored in PBS/0.1% sodium azide before microCT analysis. Immediately prior to scanning, samples were embedded in a 50:50 mix of polyester and paraffin waxes (Electron Microscopy Sciences, 19312 and 19302-01, respectively) to prevent motion artifacts and desiccation. MicroCT images were acquired by the Center for Quantitative Imaging at the Pennsylvania State University on the General Electric v|tom|x L300 nano/microCT system using the 300-kV tube at 80 kV and 180 µA using a 0.2 mm aluminum filter and image voxel size of 0.015 mm isotropic. Image data were reconstructed on a 2024×2024-pixel grid as a 32-bit volume and were reduced to 16-bit. A minimum threshold of 70-100 $mg/cm^3$ partial density hydroxyapatite (HA) based on HA phantoms imaged with the specimens was used to reconstruct three-dimensional (3D) isosurfaces of skulls for image analysis using Avizo 2021.1 (Thermo Fisher Scientific). The dataset includes 5 wild-type and 6 $Hhip^{-/-}$ samples previously reported (Holmes et al., 2021) but sample size has been increased for this analysis ($n$=11 wild type and 10 $Hhip^{-/-}$).

### Morphological evaluation of skull shape and form

The 3D coordinates of 39 biologically-relevant anatomical landmarks (Table S1) were recorded using the MALPACA method (Zhang et al., 2022) in the SlicerMorph extension of 3D Slicer. After Procrustes superimposition, a Principal Components Analysis (PCA) of the 39 anatomical landmarks describing the facial skeleton, cranial vault and cranial base of the specimens were analyzed using the General Procrustes Analysis module of SlicerMorph (Rolfe et al., 2021), an open and extensible platform to retrieve, visualize and analyze 3D morphology within 3D Slicer (Kikinis et al., 2014). Two types of PCA were performed: a PCA based on variation in form (size and shape together), followed by a PCA based on shape variation alone (corrected for allometry). Euclidean distance matrix analysis (EDMA) was used to statistically determine morphological differences between groups (Lele, 1993; Lele and Richtsmeier, 1995, 2001).

### Quantification of skull bone volumes and areas

Frontal and parietal bones were segmented from 3D isosurfaces created from microCT scans thresholded for bone of E18.5 skulls of wild-type and $Hhip^{-/-}$ embryos. Bone thickness was determined using the Surface Thickness Module of Avizo 3D 2021.2 (Thermo Fisher Scientific). Bone volume data was of normal distribution and was analyzed using paired samples t-test in IBM SPSS Statistics for Windows v. 28.0.1.0 (IBM Corp.) to determine whether left and right bone data were different. Independent samples t-tests were performed to determine if genotype affected left and right frontal and parietal bone volumes. Suture fusion was scored on microCT scans thresholded for bone as ordinal data (patent, defined as <10% bridging between bones; partially patent, defined as >10% to <90% bridging between bones; fused, defined as >90% bridging between bones).

Suture fusion data was analyzed using an Independent-Samples Kruskal–Wallis test of IBM SPSS Statistics.

## Single molecule fluorescent RNA *in situ* hybridization

Multiplexed single molecule fluorescent RNA in situ hybridization (smFISH) was performed using the RNAscope Fluorescent Multiplex Reagent Kit v1 (Advanced Cell Diagnostics, 320850) on fresh frozen sections or v2 (Advanced Cell Diagnostics, 323100) on fixed frozen sections, with the following modifications: 4% PFA fixation of sections was performed for 1 h, and Proteinase IV digestion for fresh frozen sections and Proteinase III digestion for fixed frozen sections were performed for 10 min. For the v2 kit, probes were detected with the Opal 520, 590 and 670 Reagent packs (Akoya Biosciences, FP1487001KT, FP1488001KT and FP1497001KT, respectively). Probes (Advanced Cell Diagnostics) were for *Gli1* (311001-C2), *Hhip* (448441-C3), *Ihh* (413091), *Ptch1* (402811-C2), *Pth1r* (426191), *Pthlh* (456521-C3) and *Sp7* (403401-C2). Images were acquired with an AxioImager Z2M equipped with a 20×/0.8NA Zeiss Plan-Apochromat objective, a monochrome Axiocam 503 camera (Zeiss, 1936×1460 pixels, 4.54 μm×4.54 μm per pixel, sensitivity ~400 nm–1000 nm) and Zen 2 Blue Edition software (version 2.0). Z-stack images were acquired at optimal sampling rate meeting Nyquist frequency requirements, as calculated by the software. Pseudo-colored images were made by converting grayscale images using the Color/Merge Channels function of Fiji (ImageJ2, v2.14.0/1.54f).

## Bulk RNA preparation

Coronal and lambdoid sutures were dissected in ice-cold phosphate-buffered saline (PBS) from E18.5 embryos and snap frozen on dry ice. Coronal sutures included the overlapping frontal and parietal bones with osteogenic fronts and the suture mesenchyme between them; lambdoid sutures consisted of one continuous strip that included a narrow margin of bilateral parietal and interparietal bone, and the intervening suture and posterior fontanel mesenchyme. Hypodermis and dura mater tissue were removed as far as possible. Care was taken to cleanly separate the lambdoid suture from the underlying cranial cartilage, the tectum posterius (see Fig. S3), by peeling the tissues apart with fine forceps. Four replicate wild-type and *Hhip*$^{-/-}$ libraries were generated from two male and two female embryos of each genotype, giving a total of eight libraries for each suture. Total RNA was prepared from wild-type and *Hhip*$^{-/-}$ paired coronal sutures and individual lambdoid suture strips using the Qiagen RNeasy Minikit (Qiagen, 74104) with on-column DNase digestion (Qiagen, 79254), as described by the manufacturer. Sutures were first homogenized for 40-60 s in 350 μl RLT buffer using a disposable Kontes pellet pestle (Fisher Scientific, K749521-1500) and pellet pestle motor (Kimble, 6HAZ6) in 1.5 ml Eppendorf tubes. RNA quality and concentration were determined using the High Sensitivity Tapestation 4200 (Agilent). RNA-seq libraries were prepared from 50 ng of RNA, between a RIN of 8.2 to 9.6, using the NEBNext Poly(A) Magnetic Isolation Module (New England Biolabs, E7490) and the NEBNext Ultra II RNA Library Prep Kit for Illumina (New England Biolabs, E7775). PolyA RNA was isolated from total RNA and fragmented for cDNA synthesis. After ligation to sequencing adapters, ligated fragments were size-selected through purification using the Sample Purification Beads included in the kit and underwent PCR amplification to prepare the libraries. The resulting library insert size was 200-500 bp with a median size around 300 bp. The 16 libraries were barcoded using unique dual indexing (New England Biolabs, E6440S) and pooled for sequencing on an Illumina NovaSeq 6000 instrument using standard protocols for S1 2×50 bp paired-end sequencing. Demultiplexed FASTQ files were generated using bcl-convert v3.7.5.

## Bulk RNA-seq analysis

After adapter sequences were removed using Cutadapt (Martin, 2011) and low-quality bases trimmed from the 3′ ends of reads with a quality score threshold of Q≤20 for stretches exceeding 20 bases, paired-end reads were aligned to the mouse mm10 reference genome using STAR (Dobin et al., 2013). Gene-level read counts were summarized with featureCounts (Liao et al., 2014). The resulting raw paired-end read counts were compiled into a numeric matrix with genes as rows and experiments as columns for downstream differential gene expression analysis using the Bioconductor Limma package (Ritchie et al., 2015) after applying several filtering steps to exclude lowly expressed genes.

First, gene counts were normalized to FPKM (fragments per kilobase per million mapped reads) using RSEM (Li and Dewey, 2011) in strand-specific mode with default parameters, retaining only genes with expression levels above 1 FPKM in at least 50% of the samples. Additional filtering removed genes with fewer than 50 total reads across all samples or with transcript lengths shorter than 200 nucleotides. Normalization factors were calculated for the filtered data matrix using the TMM (trimmed mean of M-values) method. Subsequently, the data underwent voom mean-variance transformation (Law et al., 2014) to prepare for Limma linear modeling, with sex included as a fixed-effect covariate. Prior to modeling, a PCA was performed on the voom transformed data to identify outliers that did not cluster with their sample groups. One outlier sample, a *Hhip*$^{-/-}$ female coronal suture, was removed during this process. A design matrix was constructed to represent the experimental groups, enabling pairwise comparisons across sample groups. Adjusted P-values were calculated using the eBayes method and corrected for multiple comparisons using the Benjamini-Hochberg (BH) procedure, with genes deemed significantly differentially expressed based on a false discovery rate (FDR) threshold of q≤0.05.

Gene Ontology (GO) and KEGG enrichment analyses of differentially expressed genes were conducted using gProfileR2 (FDR≤0.05) (Kolberg et al., 2020). Analyses were performed on ranked gene lists based on fold-change in expression, using a custom background of expressed genes (>1 FPKM in at least 50% of samples) with GO or KEGG annotations.

## Statistical analysis

Sample sizes (*n*) are given in the text or figure legends. For quantification of EdU incorporation and cell number, statistical analysis and dot plotting were performed with Microsoft Excel for Mac, version 16.16.27, and statistical significance was determined using an unpaired two-tailed Student's *t*-test. For comparison of bone volumes by micro CT, statistical analysis was performed with IBM SPSS Statistics version 28.0.1.0(142) and statistical significance was determined using the Mann–Whitney *U*-test. *P*<0.05 were considered significant.

### Acknowledgements

The authors thank the Genetic Resources Core Facility of Johns Hopkins University for RNA sequencing support, the Microscopy and Advanced Bioimaging Core of the Icahn School of Medicine at Mount Sinai for microscopy support, the Center for Biostatistics of the Icahn School of Medicine at Mount Sinai for statistical support, and the Penn State Center for Quantitative Imaging for microCT support. This work was supported in part through the computational resources and staff expertise provided by Scientific Computing at the Icahn School of Medicine at Mount Sinai. Part of Fig. 8 is reproduced in the Introduction to the Master's thesis of Q.L. (Icahn School of Medicine at Mount Sinai, 2025).

### Competing interests
The authors declare no competing or financial interests.

### Author contributions
Conceptualization: G.H.; Data curation: M.D.S., S.M.M.P., H.v.B., G.H.; Formal analysis: M.D.S., S.M.M.P., H.v.B., G.H.; Funding acquisition: J.T.R., E.W.J., H.v.B., G.H.; Investigation: M.D.S., S.M.M.P., Q.L., H.v.B., G.H.; Methodology: G.H.; Project administration: G.H.; Software: S.M.M.P., H.v.B.; Supervision: G.H.; Validation: M.D.S., S.M.MP., H.v.B., G.H.; Visualization: M.D.S., S.M.M.P., H.v.B., G.H.; Writing – original draft: S.M.M.P., G.H.; Writing – review & editing: M.D.S., S.M.M.P., J.T.R., E.W.J., H.v.B., G.H.

### Funding
This work was funded by the National Institutes of Health (R01DE030596 to E.W.J., H.v.B. and G.H.; R01DE027677 to J.T.R.). Open Access funding provided by the Icahn School of Medicine at Mount Sinai. Deposited in PMC for immediate release.

### Data and resource availability
The RNA sequencing data reported in this publication have been deposited in the NCBI Gene Expression Omnibus under accession number GSE294710. All other relevant data and details of resources can be found within the article and its supplementary information.

**Peer review history**
The peer review history is available online at https://journals.biologists.com/dev/lookup/doi/10.1242/dev.204875.reviewer-comments.pdf

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
