## [Peer Review File · Development (Cambridge, England)]

Differential regulation of coronal and lambdoid suture patency by PTHLH and HHIP activity in mice

Madrikha D. Saturne, Susan M. Motch Perrine, Qingyang Li, Joan T. Richtsmeier, Ethylin Wang Jabs, Harm van Bakel and Greg Holmes
DOI: 10.1242/dev.204875

Editor: Liz Robertson

Review timeline

Original submission:	17 April 2025
Editorial decision:	8 June 2025
First revision received:	7 August 2025
Editorial decision:	3 September 2025
Second revision received:	4 September 2025
Accepted:	5 September 2025

Original submission

First decision letter

MS ID#: dev.204875

MS TITLE: Suture-specific regulation of coronal and lambdoid suture patency by PTHLH and HHIP activity in mice

AUTHORS: Greg Holmes; Madrikha D Saturne; Susan Motch Perrine; Qingyang Li; Joan T Richtsmeier; Ethylin Wang Jabs; Harm van Bakel

Dear Dr Holmes,

I have now received all the referees reports on the above manuscript, and have reached a decision. The referees' comments are appended below, or you can access them online: please go to .

The overall evaluation is positive and we would like to publish a revised manuscript in Development, provided that the referees' comments can be satisfactorily addressed. Please attend to all of the reviewers' comments in your revised manuscript and detail them in your point-by-point response. If you do not agree with any of their criticisms or suggestions explain clearly why this is so. If it would be helpful, you are welcome to contact us to discuss your revision in greater detail. Please send us a point-by-point response indicating your plans for addressing the referees' comments, and we will look over this and provide further guidance.

Reviewer 1

SUMMARY OF THE ADVANCE MADE IN THIS PAPER AND ITS POTENTIAL SIGNIFICANCE TO THE FIELD

In syndromic craniosynostosis, calvarial sutures are variably affected, often depending on the specific disease-causing gene involved . The extent to which this variability is driven by suture-specific molecular programs remains poorly understood. This study investigates a mechanism in the coronal suture that is protective against hedgehog-related craniosynostosis in mice. Hedgehog (HH) signaling regulates osteogenic differentiation within the calvarial sutures, and enhanced HH activity

causes craniosynostosis in humans and mice. Previous work by the authors showed that loss Hhip, an inhibitor of Hedgehog signaling, alters coronal suture morphology without causing fusion. In the current study, they found that loss of Hhip induces fusion of the lambdoid suture and investigated transcriptional differences that explain these suture-specific phenotype outcomes in response to increased HH signaling. RNA-seq and smFISH revealed that the HH target genes *Gli1* and *Ptch1* are upregulated, whereas *Ihh* is paradoxically downregulated in both the coronal and lambdoid sutures of Hhip knockout mice. They also found that *Pthlh* expression, while upregulated in both the coronal and lambdoid sutures of Hhip knockout mice, exhibits relatively higher expression in the coronal. To test the idea that *Pthlh* acts as a negative regulator of *Ihh* expression that prevents fusion of the Hhip knockout coronal suture, they generated double knockout Hhip^{-/-}; Pthlh^{-/-} mice. They found extensive coronal suture fusion in Hhip^{-/-}; Pthlh^{-/-} mice, suggesting that *Pthlh* is protective against ectopic HH signaling specifically in the coronal suture. Overall, this study sheds new light on suture-specific requirements for HH signaling levels.

SUGGESTIONS TO AUTHORS

The following considerations should be addressed to strengthen clarity and the overall conclusions.

Major:

1. Move the apoptosis results to the supplemental data.
2. Move the data from Figure S4 into Figure 3 so that suture-specific differences in the pattern of gene expression can be fully appreciated. This is a fundamental aspect of the paper.
3. Indicate sample size within the results section or Figure legends where missing (gene expression analyses, RNA-seq, etc.).
4. The results section should describe how 1) total cell number was measured and 2) it was determined that the increase in cell number in the near fusion region of Hhip KO lambdoid was derived from the IP only (Figure 2).
5. In Figure 3, there is little to no lacZ in the lambdoid suture of Hhip^{+/-} mice. In the coronal suture, on the other hand, there is lacZ in Hhip^{+/-} mice (Figure 9 of Holms et al., 2021). This suggests that baseline *Ihh* signaling levels are different - with the lambdoid having none and coronal having some. Is it possible then that the lambdoid suture is more sensitive to Hhip loss because it is required to completely suppress *Ihh*? The coronal suture, on the other hand, may have a higher tolerance to *Ihh* signaling levels.
6. The RNA-seq analysis suggests that there is an increase in osteoclasts in the Hhip KO coronal and an increase in mast cells in the Hhip KO lambdoid. How are these differences expected to mechanistically relate to phenotypic differences (no fusion versus fusion, HH signaling, etc.)?
7. Line 412 - it is said that differential expression of HH pathway genes may underlie suture-specific variation in HH signaling strength. It would be very exciting to show this more directly by showing side-by-side lacZ staining Hhip^{+/-} mice and also looking at protein levels of *Gli1* and *Ihh* in coronal and lambdoid. RNA is not always a perfect readout for protein levels/localization.
8. It is concluded that the coronal suture of *Ihh* KO mice is less developed compared to control. While this is somewhat supported by the ALP stain shown, the conclusion should be strengthened by additional phenotypic analysis.
9. While the *Ihh* KO phenotype in the lambdoid suture was briefly described in whole mount skeletal preparation by Veistinen et al., 2017, it would be helpful to see the lambdoid phenotype histologically side-by-side with the coronal.
10. In Figure 7, the signal is much lower in the WT coronal compared to other WT coronal samples with the same probes in the other figures. This gives the impression that the RNA is degraded in these samples.
11. Additional phenotypic data of the Hhip/*Pthlh* double mutants (microCT or skeletal preps) to show the regionality of the phenotype would strengthen the conclusion. Since *Pthlh* is upregulated in the lambdoid of Hhip KO mice, the lambdoid suture should also be shown.
12. The coronal suture is derived from both NCC and mesoderm, whereas the lambdoid is largely mesodermal. There should be some discussion about the extent to which the mixed embryonic origin may have an impact on suture-specific sensitivity to changes in HH signaling in the discussion.

Minor:

1. State that Hhip is a target of HH signaling and therefore the lacZ is a readout for HH activity (line 258).

2. Line 295 - clarify the statement "no more than approximately 50% of WT levels". Does this mean that the expression levels of these genes were reduced by 50% compared to WT?
3. Lines 464-471 - the description of these observations should be made clearer.

Reviewer 2

SUMMARY OF THE ADVANCE MADE IN THIS PAPER AND ITS POTENTIAL SIGNIFICANCE TO THE FIELD

This paper makes an original contribution to the field of craniofacial, developmental biology. The role of HH signalling in the regulation of suture development and patency is not understood fully and the data presented in this paper make a big step forwards. This will be interesting to craniofacial biologists and clinical scientists and is likely to be cited widely if accepted for publication.

SUGGESTIONS TO AUTHORS

Title

* There seems to be some unnecessary duplication by saying 'suture-specific' and 'suture patency'. I feel that the title is fine without 'suture-specific' and shorter is always better.

Introduction

No comments.

Results

* I don't understand why the results section starts with the non-cranial phenotypes and all the data is in a supplementary figure. If this is relevant, this should be Figure 1, if it is not, there is no need to include this data. I would prefer the latter. The relevant section in the Discussion can also be removed.

* It seems likely that the increased body weight is related to the increased skull size, but the authors don't make this explicit. If this is indeed the case, please add this or make it clear that there is a general overgrowth instead.

* The micro-CT data suggests that some cranial bones are larger and/or thicker. Can the authors comment on the possibility that bone density has changed/increased?

* The interpretation of in situ expression data is not satisfactory. This applies mainly to Figure 3. ISH is by no means a quantitative method. Ectopic expression can be detected more confidently. Furthermore, using data from fused bones should be done with care, as the changed anatomy and lack of SM can affect the interpretation of the expression pattern. My analysis based on Figure 3 is that 1) expression of *lhh* is decreased in the OF, 2) expression of HH downstream markers (*Gli1* and *Ptch1*) is not significantly different in the OF, but there is an enlarged expression domain in the SM. This makes sense as *Hhip* is expressed in the SM and not the OF. The ectopic expression could be a prelude to fusion as it is clear that there is significant expression in the ecto- and endocranial mesenchyme in the fused samples.

In this light, the conclusion that "Taken together, these results show that in both sutures, HH transcriptional outputs are increased throughout the *Hhip*^{-/-} SM compared to WT, even as *lhh* expression is decreased." should be qualified by adding that: *lhh* expression in the OF is decreased. The question remains -for now-, why is *lhh* expression in the OF decreased?

Line 247- *Hhip* expression in the lambdoid suture was enriched in the OFs and adjacent SM, similar to *Gli1* and *Ptch1* (Fig. 3A). I suggest using 'expressed' instead of 'enriched' here. This sentence only describes the expression pattern in WT, there is no comparison, which 'enriched' suggests.

Line 252- Surprisingly, in the *Hhip*^{-/-} lambdoid suture the domain and intensity of *lhh* expression clearly was decreased in the OFs of unfused regions compared to WT (Fig. 3B), I suggest adding the term 'unfused' or 'near fusion', to more clearly indicate this relates to panel 3B only.

Line 254- *lhh* expression was high in osteoblasts How do you know the expression is in osteoblasts without identifying where the osteoblast are? Osteocytes maybe?

* The *Pthlh* experiments identifying an association with *lhh* and offering an explanation for the decrease in *lhh* in the OF are excellent. An outstanding contribution to the understanding of cranial development and the role of HH signalling!

Discussion

* The Discussion about the observed overgrowth phenotype is unclear (lines 540-549). Are the authors claiming that this is mainly caused by the ectopic bone formation at the lower posterior margin of the parietal bone? It seems to suggest that. It would be good -as mentioned above- to be

able to understand if the increase in size is whole body, skeletal-only, cranial-only or a combination.

* In the section about how PTHLH regulates HH (line 571-onwards) some comment or speculation about a role for FGF signalling would make the discussion more interesting. FGF signalling is the major pathway in cranial (suture) development and many genes have been associated with craniosynostosis. There is some evidence that FGF and HH signalling interact and regulate each other in different contexts, including craniofacial tissues.

* The conclusions from this study are robust, but complex. The paper would greatly benefit from a graphical summary of the proposed mechanism.

Methods

* The detailed description of the suture dissection for RNAseq is very much appreciated. This is often missing in similar papers, but very important.

First revision

Author response to reviewers' comments

Comments from the Reviewers:

Note: Text revisions are highlighted in yellow in the manuscript and indicated with revised line numbers in the individual responses below.

Reviewer 1: SUMMARY OF THE ADVANCE MADE IN THIS PAPER AND ITS POTENTIAL SIGNIFICANCE TO THE FIELD

In syndromic craniosynostosis, calvarial sutures are variably affected, often depending on the specific disease-causing gene involved. The extent to which this variability is driven by suture-specific molecular programs remains poorly understood. This study investigates a mechanism in the coronal suture that is protective against hedgehog-related craniosynostosis in mice. Hedgehog (HH) signaling regulates osteogenic differentiation within the calvarial sutures, and enhanced HH activity causes craniosynostosis in humans and mice. Previous work by the authors showed that loss Hhip, an inhibitor of Hedgehog signaling, alters coronal suture morphology without causing fusion. In the current study, they found that loss of Hhip induces fusion of the lambdoid suture and investigated transcriptional differences that explain these suture-specific phenotype outcomes in response to increased HH signaling. RNA-seq and smFISH revealed that the HH target genes Gli1 and Ptch1 are upregulated, whereas Ihh is paradoxically downregulated in both the coronal and lambdoid sutures of Hhip knockout mice. They also found that Pthlh expression, while upregulated in both the coronal and lambdoid sutures of Hhip knockout mice, exhibits relatively higher expression in the coronal. To test the idea that Pthlh acts as a negative regulator of Ihh expression that prevents fusion of the Hhip knockout coronal suture, they generated double knockout Hhip-/-;Pthlh-/- mice. They found extensive coronal suture fusion in Hhip-/-;Pthlh-/- mice, suggesting that Pthlh is protective against ectopic HH signaling specifically in the coronal suture. Overall, this study sheds new light on suture-specific requirements for HH signaling levels.

SUGGESTIONS TO AUTHORS

The following considerations should be addressed to strengthen clarity and the overall conclusions. Major:

1. *Move the apoptosis results to the supplemental data.*

We now present these data (previously Fig 2G) as a new supplementary Fig S1. The text in Results has been changed accordingly.

2. Move the data from Figure S4 into Figure 3 so that suture-specific differences in the pattern of gene expression can be fully appreciated. This is a fundamental aspect of the paper.

These data now appear in a revised Figure 3 as panels D and E, and the text was altered accordingly.

3. Indicate sample size within the results section or Figure legends where missing (gene expression analyses, RNA-seq, etc.).

Sample sizes are now indicated in all Figure and Supplemental Figure legends, as well as in the text, where relevant.

4. The results section should describe how 1) total cell number was measured and 2) it was determined that the increase in cell number in the near fusion region of Hhip KO lambdoid was derived from the IP only (Figure 2).

We have addressed this by making changes in both the Results and the Materials and Methods sections (“Immunohistochemistry and cytochemistry”) to make clearer that EdU and total cell number were determined individually for the different osteogenic fronts and the suture mesenchyme, as described in the original Methods text.

In the Results we have added or modified the following text (new lines 207-213) (changes highlighted):

Changes in cell proliferation were determined by quantifying EdU incorporation as a percentage of total cell number in individual parietal and interparietal OFs and intervening SM. Proliferation in the OFs and SM did not differ between WT and unfused regions of *Hhip*^{-/-} lambdoid sutures (Fig. 2D,E). There was a significant decrease in total cell number in the *Hhip*^{-/-} SM, reflecting a shortening of the mutant SM near fused regions, and a small but significant increase in cell numbers in the interparietal OF...

In the Materials and Methods we have added the following clarifying text (new lines 664- 667) (changes highlighted):

Sections were counterstained with DAPI. EdU-positive nuclei and total nuclei were counted within the individual OFs and SM of five non-consecutive sections per lambdoid suture and averaged. In *Hhip*^{-/-} sutures only unfused regions adjacent to fused regions were assessed. In OFs nuclei...

5. In Figure 3, there is little to no lacZ in the lambdoid suture of *Hhip*^{+/-} mice. In the coronal suture, on the other hand, there is lacZ in *Hhip*^{+/-} mice (Figure 9 of Holmes et al., 2021). This suggests that baseline *Ihh* signaling levels are different - with the lambdoid having none and coronal having some. Is it possible then that the lambdoid suture is more sensitive to *Hhip* loss because it is required to completely suppress *Ihh*? The coronal suture, on the other hand, may have a higher tolerance to *Ihh* signaling levels.

The Reviewer raises an interesting point about why the coronal and lambdoid sutures have different responses to *Hhip* deletion, although we would say our data demonstrate clear HH signaling activity in the lambdoid suture, rather than none. We would first point out that the LacZ reporter is an indicator of HH signaling strength specifically at the genetically-modified *Hhip* locus. Our Table 1 shows that endogenous WT *Hhip* expression is approximately twice as high in the coronal suture compared to the lambdoid, and by RNAscope is physically more concentrated in the coronal suture mesenchyme compared to an apparently more diffuse expression along the lambdoid osteogenic fronts. The very low levels of LacZ signal detected in the *Hhip*^{+/-} lambdoid suture may therefore reflect a combination of these factors and the sensitivity of the LacZ assay itself, which facilitates detection of signaling in the coronal suture. As in our Fig 9 from Holmes et al, 2021, LacZ detection appears strongest in the most

central region of Hhip expression detected by RNAscope, rather than throughout the entire domain of Hhip expression in the suture mesenchyme and osteogenic fronts. Second, our Table 1 suggests that by some measures the baseline Ihh signaling level is comparable between the WT coronal and lambdoid sutures, as the levels of the HH transcriptional readouts, Gli1 and Ptch1, are not significantly different between them (new line 406). The RNAscope images also clearly show readily detectable HH target gene expression (Gli1, Ptch1, Hhip) in the lambdoid suture, showing active HH signaling. Finally, we would point out that Hhip is not the only HH inhibitor required for maintaining an open lambdoid suture. The Ptch1 hypomorph mutant also causes lambdoid suture fusion preferentially, despite the presence of Hhip. We reference this hypomorph phenotype in new lines 114 and 530-532, with the suggestion that “the murine lambdoid suture may be more susceptible to fusion when HH signaling is increased” (new lines 532-534). This susceptibility may be due to differential expression of the variety of HH-related genes specifically listed in Table 1, which includes genes for proteins that act at different steps of the HH ligand secretion, transport, and reception, and for Pthlh, as well as the differing structure of the two sutures, as we suggest in the Results (new lines 404-414) and Discussion (new lines 512-518 and 535-545).

In our view, these points are already consistent with the Reviewer’s suggestion, but based on the finding of active HH signaling in both sutures. In our phrasing, the lambdoid suture is more sensitive to loss of Hhip because in the lambdoid suture it is a major effective restrictor of Ihh signaling (along with Ptch1), whereas the coronal suture additionally has high levels of Pthlh that partially compensate for loss of Hhip. As we believe we have provided extensive discussion of these issues in the text, we have not added additional text to further this point.

6. The RNA-seq analysis suggests that there is an increase in osteoclasts in the Hhip KO coronal and an increase in mast cells in the Hhip KO lambdoid. How are these differences expected to mechanistically relate to phenotypic differences (no fusion versus fusion, HH signaling, etc.)?

This may be a misinterpretation of our text under the section “Comparison of transcriptome changes between Hhip^{-/-} coronal and lambdoid sutures”. In this section we also compare the WT coronal and lambdoid transcriptomes, and state: “Overall, these results show that the coronal and lambdoid sutures differ ... with osteoclasts enriched in the coronal suture and mast cells enriched in the lambdoid suture.” (new lines 400-403). This is the only comparison where mast cell markers differ, and their expression is not changed by loss of Hhip in either suture. While there is literature supporting a role for mast cells in pathologies of mature bone, we have seen no studies suggesting a role in embryonic bone formation. Our own, unrelated assay of mast cell distribution at this age suggests they are sparse over the bone, and therefore are more likely to be found in a suture with a larger area, such as the lambdoid, than in a suture with a smaller area, such as the coronal. As we state, osteoclast markers are more highly expressed in the WT coronal suture compared to the WT lambdoid suture, and are actually decreased by about 50% in the Hhip KO coronal suture. On the other hand, osteoclast marker expression does go up highly in the Hhip KO lambdoid suture, and we did suggest a possible connection with this to the correlated upregulation of osteoclast inducers Rankl and Csf1 that are expressed by osteoblasts (Results, new lines 341-342 and Discussion, new lines 574-578). In various mouse models of embryonic craniosynostosis, osteoclast activity has not been described as a causative factor for fusion, and we would suggest that in the Hhip KO the increase in osteoclast marker expression, and presumably of osteoclast numbers, is secondary to increased ossification.

To address this comment and better differentiate the description of differences between the WT coronal and lambdoid suture from that of the Hhip KO sutures, we have created the new section heading “Transcriptome differences between WT coronal and lambdoid sutures”. (new line 384).

7. Line 412 - it is said that differential expression of HH pathway genes may underlie suture-specific variation in HH signaling strength. It would be very exciting to show this more directly by showing side-by-side lacZ staining Hhip^{+/-} mice and also looking at protein levels of Gli1 and Ihh in coronal and lambdoid. RNA is not always a perfect readout for protein

levels/localization.

We would suggest that expression levels of each of the HH target genes (Gli1, Ptch1, and Hhip) between WT coronal and lambdoid sutures (as in Fig. 3 and Table 1) provide equally interesting comparisons of HH signaling strength to LacZ staining, which is essentially another target gene read-out. For the reasons we discuss in Point 5 above, the LacZ read-out alone does not represent the sum of HH signaling in general. We would point out here that Hhip expression in the coronal suture is spatially distinct from Gli1 and Ptch1, even though they are all HH target genes. This suggests further complexity in the relationship between HH signaling and target gene regulation, at least in the coronal suture. However, we have provided LacZ staining images for the Hhip heterozygote and knockout coronal sutures (similar to our data in Holmes et al, 2021) to allow further appreciation of the difference between lambdoid and coronal sutures within Fig. 3. Additional text has been included (new lines 249-254).

We agree that protein visualization would be a valuable addition to any study of HH signaling in the sutures. However, we have conducted immunohistochemistry with various commercial antibodies for Ihh, Gli1, and Hhip, with various protocols, and have not been able to obtain clear signals in any case. While the RNA may not accurately reflect protein levels/localization, the expression levels of Gli1, Ptch1, and Hhip do provide a readout of HH signaling strength. The lack of protein expression data does not alter the finding of this study, which is the supportive role of Pthlh in maintaining coronal suture patency in the absence of Hhip.

8. It is concluded that the coronal suture of Ihh KO mice is less developed compared to control. While this is somewhat supported by the ALP stain shown, the conclusion should be strengthened by additional phenotypic analysis.

Points 8 and 9 relate to the Ihh KO phenotype, and are answered here. We now include a new supplemental Fig. S3 to show typical sagittal sections through WT and Ihh^{-/-} calvaria with enlarged images of the corresponding coronal and lambdoid (Point 9) sutures, stained for alkaline phosphatase (ALP), which clearly show the under-developed aspect of the KO sutures. We include additional text and references to previous publications that specifically show coronal and lambdoid defects in calvaria of Ihh^{-/-} embryos. As we use the Ihh KO model to show the dependence of Pthlh expression, and the Ihh phenotype itself is not the focus of the paper, we trust this characterization is sufficient to make our point. The revised section is as follows (new lines 465-477) (changes highlighted):

In the growth plate, *Pthlh* expression depends on the expression of *Ihh* (St-Jacques et al., 1999). *IHH* also induces *Pthlh* expression in mature, postnatal osteoblasts (Mak et al., 2008). In *Ihh*^{-/-} embryos, coronal and lambdoid sutures appear wider in whole calvaria stained for mineralized bone (Abzhanov et al., 2007; Klopocki et al., 2011; Veistinen et al., 2017). We confirmed by histological sectioning that E18.5 *Ihh*^{-/-} coronal and lambdoid suture development was impaired compared to WT (Fig. 7 and Fig. S3). Mutant coronal sutures typically lacked the overlap between frontal and parietal bones (Fig. 7A,B and Fig. S3A,B,D). Mutant OFs in both sutures were also thinner than WT, and ALP activity was weaker throughout the calvarial bones (Fig. S3), in agreement with a previous report of decreased *Alp* expression (Lenton et al., 2011). Of the known transcriptional targets of HH signaling, *Gli1*, *Hhip*, and *Ptch1* expression was absent in *Ihh*^{-/-} coronal sutures (Fig. 7C-F). *Pthlh* expression also was not detected compared to WT (Fig. 7E,F), indicating that its expression depends on *Ihh* transcription.

9. While the Ihh KO phenotype in the lambdoid suture was briefly described in whole mount skeletal preparation by Veistinen et al., 2017, it would be helpful to see the lambdoid phenotype histologically side-by-side with the coronal.

See point 8 above.

10. In Figure 7, the signal is much lower in the WT coronal compared to other WT coronal samples with the same probes in the other figures. This gives the impression that the RNA

is degraded in these samples.

We have replaced Figure 7 with a revised figure showing stronger staining and a more clearly aberrant suture development. In repeating this experiment, we included additional independent samples of WT and *Ihh* KO sutures and have increased the n values accordingly. Text changes are given in the response to point 8 above.

11. Additional phenotypic data of the *Hhip*/*Pthlh* double mutants (microCT or skeletal preps) to show the regionality of the phenotype would strengthen the conclusion. Since *Pthlh* is upregulated in the lambdoid of *Hhip* KO mice, the lambdoid suture should also be shown.

By “regionality”, we understand the Reviewer to mean the extent of the fusion within a suture. The problem with microCT or skeletal staining preparations for this purpose is that they image the mineralized portion of the skeleton and not the unmineralized osteoid. Deposition of unmineralized osteoid joining the frontal and parietal bones, or the parietal and interparietal bones, is the initial fusion event in each suture, and is ideally seen by histological sectioning. This difference in results by methodology is evident in our results for lambdoid suture fusion in the *Hhip*^{-/-} embryos, where mineralized fusion was evident in 3/10 embryos using microCT, but osteoid deposition and fusion was seen at least unilaterally in 10/10 *Hhip*^{-/-} embryos using histology.

To address the Reviewer’s request, we have quantified the extent of coronal and lambdoid fusion in our histological sections stained for alkaline phosphatase, which is expressed in osteoblasts from the earliest stages of differentiation. Essentially, we have determined the percentage of sections showing definitive fusion, as shown by the presence of continuous osteoid between bone pairs of each suture, compared to the total number of sections along the length of each suture. A more detailed description has been added as a new section to the Materials and Methods (“Histological determination of the extent of suture fusion”, new lines 693-713). As we now quantify the degree of fusion, we have changed the description of coronal suture fusion from “extensive bilateral (5/6) or unilateral (1/6) fusion” to “varied degrees of bilateral (6/6) fusion”. We also include quantification of lambdoid suture fusion, which is actually increased in the double KO compared to the single *Hhip* KO. This finding is a valuable observation, as it broadens the significance of the *Pthlh*/*Ihh* interaction beyond the coronal suture, while still illustrating the differential contribution of this interaction in maintaining open coronal and lambdoid sutures. We thank the Reviewer for prompting this observation. The revised results are described in new lines 482-489 (changes highlighted):

In contrast to the *Hhip*^{-/-} coronal suture, *Hhip*^{-/-};*Pthlh*^{-/-} coronal sutures showed varied degrees of bilateral (6/6) fusion (Fig. 8C-E). Fusion, determined by the presence of continuous osteoid between frontal and parietal sutures in histological sections, spanned approximately 49% (±SD20%; n=12) of the length of individual coronal sutures. Deletion of both genes was required for fusion by E18.5, as *Hhip*^{+/-};*Pthlh*^{-/-} and *Hhip*^{-/-};*Pthlh*^{+/-} sutures did not fuse (Fig. S4). *Hhip*^{-/-};*Pthlh*^{-/-} lambdoid sutures also showed significantly more fusion than *Hhip*^{-/-} lambdoid sutures (49±SD14%, n=12, compared to 33±SD12%, n=8; t-test, p=0.018).

Given the identification of increased lambdoid suture fusion in the double KO, we have added the following line in the first paragraph of the discussion (new lines 521-523):

Loss of *Pthlh* also augments the extent of *Hhip*^{-/-} lambdoid suture fusion, suggesting a similar but much less critical role in the lambdoid suture.

12. The coronal suture is derived from both NCC and mesoderm, whereas the lambdoid is largely mesodermal. There should be some discussion about the extent to which the mixed embryonic origin may have an impact on suture-specific sensitivity to changes in HH signaling in the discussion.

We have added the following paragraph in the Discussion (new lines 546-568):

The coronal and lambdoid sutures also differ in their relative contributions of neural crest and mesoderm. In the coronal suture the frontal bone is derived from neural crest, while the SM and parietal bone are derived from mesoderm; in the lambdoid suture the parietal bone and SM are derived from mesoderm, while the interparietal bone is derived mainly from mesoderm but with a central portion derived from neural crest (Jiang et al., 2002; Yoshida et al., 2008). The lateral regions of the lambdoid suture fused at E18.5 in our study are derived from mesoderm. The potential contribution of cell lineage on the differing sensitivity to loss of *Hhip* of the two sutures is unclear. Osteoblasts from frontal bones have been characterized as having greater proliferative, osteogenic, and regenerative capacities compared to parietal osteoblasts, correlating with differences in FGF, BMP, WNT, and TGF β signaling (Doro et al., 2019; Li et al., 2010; Menon et al., 2021). A few studies have defined global transcriptional differences between frontal and parietal bones, but it is unclear whether these are lineage-specific or bone-specific differences (Chen et al., 2019; Homayounfar et al., 2015). However, the two critical genes that are the focus of our study, *Hhip* and *Pthlh*, are principally expressed in mesoderm-derived tissues in both sutures, while being more highly expressed in the coronal suture (Table 1), and so differences in cell lineage between the sutures do not appear to be a factor in their differing response to loss of *Hhip*. Gene expression differences also may reflect more the differing structure of the two sutures. However, the lack of fusion in neural crest-derived *Hhip*^{-/-} facial sutures with either overlapping or end-to-end structures (Fig. S2) suggests that neither cell lineage nor suture structure are key determinants of susceptibility to suture fusion in the absence of *Hhip*.

Minor:

1. State that *Hhip* is a target of HH signaling and therefore the *lacZ* is a readout for HH activity (line 258).

We have added the following text (new lines 243-245) (changes highlighted):

The *Hhip*^{-/-} mouse incorporates a LacZ reporter driven by the *Hhip* promoter. As *Hhip* is a target of HH signaling, LacZ expression provides a readout of HH activity at the *Hhip* locus (Chuang and McMahon, 1999).

2. Line 295 - clarify the statement "no more than approximately 50% of WT levels". Does this mean that the expression levels of these genes were reduced by 50% compared to WT?

Yes, but some of the genes named were expressed at 60 or 70% of WT levels, hence our use of "no more than", which we thought indicated that expression levels of some genes were closer to WT than 50%. However, to avoid ambiguity we have changed the sentence as indicated by the highlighted text (new lines 283-285):

The bone markers included *Bglap*, *Bglap2*, *Ibsp*, and *Ifitm5*, which were expressed at 48-71% of WT levels (Table S3).

3. Lines 464-471 - the description of these observations should be made clearer.

We have revised this paragraph as follows, and as indicated by the highlighted text (new lines 453-464):

To determine whether these changes correlated with a shift in the relative location of *Ihh* expression, we performed smFISH for *Ihh*, *Sp7* (first expressed in committed preosteoblasts), and *Pthlh* (Fig. 6). In the WT E18.5 coronal and lambdoid suture OFs, *Ihh* expression began in the OFs immediately behind the start of the *Sp7* expression domain (Fig. 6A,C). In the *Hhip*^{-/-} coronal and lambdoid suture OFs, decreased *Ihh* expression was less consistently close to the start of *Sp7* expression, but was not always displaced (Fig.

6B,D). Where the *Hhip*^{-/-} lambdoid suture fused, both *Sp7* and *Ihh* were expressed throughout the bone (Fig. 6E). These results suggest that the increased and shifted (in the coronal suture) *Pthlh* expression did not displace the *Ihh*-expressing population relative to the *Sp7*-expressing population, but specifically downregulated *Ihh* expression and/or altered the differentiation state of osteoblasts permissive for *Ihh* expression in the OF.

Reviewer 2: SUMMARY OF THE ADVANCE MADE IN THIS PAPER AND ITS POTENTIAL SIGNIFICANCE TO THE FIELD

This paper makes an original contribution to the field of craniofacial, developmental biology. The role of HH signalling in the regulation of suture development and patency is not understood fully and the data presented in this paper make a big step forwards. This will be interesting to craniofacial biologists and clinical scientists and is likely to be cited widely if accepted for publication.

SUGGESTIONS TO AUTHORS

Title

** There seems to be some unnecessary duplication by saying 'suture-specific' and 'suture patency'. I feel that the title is fine without 'suture-specific' and shorter is always better.*

We recognize the potential redundancy of “suture” appearing twice in the title, but feel it is critical to convey a difference between the two sutures in the title. We believe the following revised title reduces the duplication and length while retaining our intended meaning (change highlighted):

Differential regulation of coronal and lambdoid suture patency by PTHLH and HHIP activity in mice

Introduction No comments.

Results

** I don't understand why the results section starts with the non-cranial phenotypes and all the data is in a supplementary figure. If this is relevant, this should be Figure 1, if it is not, there is no need to include this data. I would prefer the latter. The relevant section in the Discussion can also be removed.*

These data have been removed from the text and figures. To streamline the new transitions we have added the following introductory text: 1) to the beginning of the Results section (new lines 151-153), and 2) to the second paragraph of the Discussion (new lines 525-527) (changes highlighted):

1) We previously described coronal suture dysgenesis, with the loss of SM and close approximation of frontal and parietal bones, in *Hhip*^{-/-} embryos (Holmes et al., 2021). Here, we further investigated changes in the skull of *Hhip*^{-/-} embryos.

2) Increased bone growth was evident in the skull of *Hhip*^{-/-} embryos, the size of which was larger overall, with the frontal and parietal bones having a greater volume and thickness compared to WT. Additionally, ectopic bone...

** It seems likely that the increased body weight is related to the increased skull size, but the authors don't make this explicit. If this is indeed the case, please add this or make it clear that there is a general overgrowth instead.*

We believe the overgrowth is a general phenomenon. Increases in skull distances in the mutants are between 5-15%, which would not account for the average body weight increase of 20%.

However, removal of the non-cranial data renders the question moot.

* *The micro-CT data suggests that some cranial bones are larger and/or thicker. Can the authors comment on the possibility that bone density has changed/increased?*

We have added the following text (new lines 170-173):

The mutant areas of greatest bone thickness (red in Fig. 1F,G), as determined by increased pixel density, also mapped to areas that showed increased bone mineral density, as determined by higher pixel intensity along the calibration curve defined by a co-scanned hydroxyapatite phantom.

* *The interpretation of in situ expression data is not satisfactory. This applies mainly to Figure 3. ISH is by no means a quantitative method. Ectopic expression can be detected more confidently. Furthermore, using data from fused bones should be done with care, as the changed anatomy and lack of SM can affect the interpretation of the expression pattern.*

*My analysis based on Figure 3 is that 1) expression of *Ihh* is decreased in the OF, 2) expression of HH downstream markers (*Gli1* and *Ptch1*) is not significantly different in the OF, but there is an enlarged expression domain in the SM. This makes sense as *Hhip* is expressed in the SM and not the OF. The ectopic expression could be a prelude to fusion as it is clear that there is significant expression in the ecto- and endocranial mesenchyme in the fused samples.*

*In this light, the conclusion that "Taken together, these results show that in both sutures, HH transcriptional outputs are increased throughout the *Hhip*^{-/-} SM compared to WT, even as *Ihh* expression is decreased." should be qualified by adding that: *Ihh* expression in the OF is decreased. The question remains -for now-, why is *Ihh* expression in the OF decreased?*

We agree that ISH is not quantitative and better suited to identify ectopic or expanded expression, which we meant to convey in our description of Fig. 3. The ISH data are complemented by the LacZ data, which while not quantitative do show a clear increase in signaling at the *Hhip* locus. In this light we think our description does correspond to that of the reviewer, with our emphasis of the extended SM expression of *Ihh* target genes in the mutant SM. We would point out that the *Hhip* expression domain in the lambdoid suture does include the OFs, as it overlaps with the dashed outlines of each bone, as also seen for *Gli1* and *Ptch1*. We also included the unfused regions of *Hhip*^{-/-} sutures specifically because, as the reviewer states, they should demonstrate the changes that preceded fusion. Fused regions are also important to show, as they are the key phenotype reported and show intrinsically informative expression changes compared to the "pre-fusion" regions.

We have modified the concluding sentence (new lines 254-256) (changes highlighted):

Taken together, these results show that in both sutures, HH transcriptional outputs are increased throughout the *Hhip*^{-/-} SM compared to WT, even as *Ihh* expression in the OFs is decreased.

We have not included the "The question remains..." suggestion (if that was meant to be included) as the reduction of *Ihh* expression is not the only question raised by our data, and we have rearranged the initial sentences in the first paragraph of the following section, where these questions are addressed (new lines 259-260) (changes highlighted):

Reduction of *Ihh* expression in *Hhip*^{-/-} coronal and lambdoid sutures was unexpected. For example, in the *Hhip*^{-/-} embryo...

This section and the associated Fig. 3 have also been modified to include the coronal ISH originally in Fig. S4, as suggested by Reviewer 1 (Major point 2). We have also revised the title of Fig. 3 (changes highlighted):

Fig. 3. *Ihh* expression is decreased in osteogenic fronts but HH signaling is increased throughout the suture mesenchyme in E18.5 *Hhip*^{-/-} lambdoid and coronal sutures.

Line 247- Hhip expression in the lambdoid suture was enriched in the OFs and adjacent SM, similar to Gli1 and Ptch1 (Fig. 3A). I suggest using 'expressed' instead of 'enriched' here. This sentence only describes the expression pattern in WT, there is no comparison, which 'enriched' suggests.

This and a similar change (due to the inclusion of the Fig S4 coronal data in the revised Figure 3) have been made (highlighted) (new lines 232-235):

Hhip was expressed in the OFs and adjacent SM, similar to *Gli1* and *Ptch1* (Fig. 3A). This differed from the coronal suture, where expression extends from the OFs but is highest throughout the SM between the overlapping frontal and parietal bones (Fig. 3D) (Holmes et al., 2021).

Line 252- Surprisingly, in the Hhip-/- lambdoid suture the domain and intensity of Ihh expression clearly was decreased in the OFs of unfused regions compared to WT (Fig. 3B), I suggest adding the term 'unfused' or 'near fusion', to more clearly indicate this relates to panel 3B only.

It is not clear where the requested change should be made, as we do use “unfused regions” in the sentence, and include the specific reference to Fig. 3B. (new lines 235- 238).

Line 254- Ihh expression was high in osteoblasts How do you know the expression is in osteoblasts without identifying where the osteoblast are? Osteocytes maybe?

At this age the osteoid is thin and lacks osteocytes in the fused regions. Osteoblasts can be identified by the position of alkaline phosphatase activity in adjacent sections. Co-staining of Sp7 and *Ihh* by ISH, such as in Fig. 6, show that *Ihh* in the suture is expressed in osteoblasts.

** The Pthlh experiments identifying an association with Ihh and offering an explanation for the decrease in Ihh in the OF are excellent. An outstanding contribution to the understanding of cranial development and the role of HH signalling!*

We thank the Reviewer for this assessment of our results.

Discussion

** The Discussion about the observed overgrowth phenotype is unclear (lines 540-549). Are the authors claiming that this is mainly caused by the ectopic bone formation at the lower posterior margin of the parietal bone? It seems to suggest that. It would be good - as mentioned above- to be able to understand if the increase in size is whole body, skeletal-only, cranial-only or a combination.*

Weight data and the corresponding Discussion text on overgrowth have been removed from the manuscript. Our intended message was that the *Hhip* KO embryos showed an increase in size of the whole body (about 20% heavier than WT at E18.5, on average), but in the absence of crown-rump length data and post-axial skeletal imaging we would be unable to distinguish between the Reviewer’s suggested alternatives adequately. We do not suggest that the ectopic parietal bone could account for the weight difference as it would be far too small an amount of extra bone, and in our Results we demonstrate that the volume and size of the frontal bone (like the parietal bone) increases in the mutants in the absence of ectopic bone growth. Please also see the related comment above (“* It seems likely that the increased body weight...”).

To limit our comments to cranial bone in the Discussion we have revised the following text (new lines 525-528) (changes highlighted):

Increased bone growth was evident in the skull of *Hhip*^{-/-} embryos, the size of which was

larger overall, with the **frontal and parietal** bones having a greater volume and thickness compared to WT. **Additionally**, ectopic bone formation occurred at the lower posterior margin of the parietal bone...

** In the section about how PTHLH regulates HH (line 571-onwards) some comment or speculation about a role for FGF signalling would make the discussion more interesting. FGF signalling is the major pathway in cranial (suture) development and many genes have been associated with craniosynostosis. There is some evidence that FGF and HH signalling interact and regulate each other in different contexts, including craniofacial tissues.*

Excess activity in multiple signaling pathways can lead to craniosynostosis, and HH signaling can influence various such pathways. In Hhip mutants the range of sutures affected, and the severity of fusion, is much less compared to the major craniosynostosis syndrome mutations in mouse models. Additionally, in our RNA-seq analyses of the Hhip mutant coronal and lambdoid sutures we also see no significant changes in FGF ligand or receptor expression, apart from a decrease in *Fgf10* expression, which on its own does not offer a ready explanation for the phenotypes. However, to broaden the discussion of how aberrant HH signaling may impact other relevant pathways, we have added the following paragraph after the one referred to by the Reviewer (new lines 607-618):

Increased HH signaling resulting from loss of *Hhip* and *Pthlh* expression could promote osteogenesis in the suture through additional pro-osteogenic signaling pathways that regulate suture development. IHH promotes the expression and activity of RUNX2 at the OFs (Veistinen et al., 2017). A study of the postnatal posterior frontal suture showed that RUNX2 positively regulates the expression of effector genes of the HH, FGF, WNT, and PTH pathways (Qin et al., 2019). In turn, these pathways can positively regulate RUNX2 (Komori, 2024; Teplyuk et al., 2009). In particular, aberrant FGF signaling caused by activating mutations of FGF receptors (FGFRs), which are expressed at the OFs, is a major cause of syndromic CS (Heuze et al., 2014; Wilkie et al., 2017). However, our RNA-seq analyses of *Hhip*^{-/-} sutures did not show significant changes in FGF ligand or receptor expression, apart from a decrease in *Fgf10* expression in both sutures (Table S4).

** The conclusions from this study are robust, but complex. The paper would greatly benefit from a graphical summary of the proposed mechanism.*

We have added a new Fig. 9 to graphically summarize our proposed mechanism of coronal suture regulation by Hhip and Pthlh. It is referred to in the final sentence of the Results (new line 498, and legend between new lines 1291-1302).

Methods

** The detailed description of the suture dissection for RNAseq is very much appreciated. This is often missing in similar papers, but very important.*

We definitely agree, and thank the reviewer for their appreciation.

Additional minor grammatical changes, typo corrections (for example, new line 1224), or further additions to Materials and Methods (for example, new lines 686-691) have been made and highlighted where non-trivial.

Second decision letter

MS ID#: dev.204875R1

MS TITLE: Differential regulation of coronal and lambdoid suture patency by PTHLH and HHIP activity in mice

AUTHORS: Greg Holmes; Madrikha D Saturne; Susan Motch Perrine; Qingyang Li; Joan T Richtsmeier; Ethylin Wang Jabs; Harm van Bakel

Dear Dr Holmes,

I have now received all the referees reports on the above manuscript, and have reached a decision. The referees' comments are appended below, or you can access them online: please go to View Reviewer Comments.

The overall evaluation is positive and we would like to publish a revised manuscript in Development. However you will see that the Reviewer has made 2 very small but potentially helpful suggestions that you might want to incorporate into the final version. Your manuscript will not require any further review, rather I will accept it once you upload the final files.

Reviewer 1

This is a much improved version of the manuscript.

I have two suggestions to be considered by the authors and editor:

1. The conclusion that the domain and intensity of *Ihh* expression is increased in the *Hhip/Pthlh* double knockout compared to the *Hhip* knockout (Figure 8) is difficult to appreciate in the images. Either modify this conclusion or present more compelling data.
2. It would be helpful to include the lambdoid suture in the model figure to underscore the suture-specific roles for HH inhibitors.

Second revision

Author response to reviewers' comments

Dear Dr. Robertson,

We are thankful to submit final revisions for our manuscript, “**Differential regulation of coronal and lambdoid suture patency by PTHLH and HHIP activity in mice**”.

We have addressed Reviewer 1’s suggestions as follows:

*1. The conclusion that the domain and intensity of *Ihh* expression is increased in the *Hhip/Pthlh* double knockout compared to the *Hhip* knockout (Figure 8) is difficult to appreciate in the images. Either modify this conclusion or present more compelling data.*

This conclusion was stated twice, in lines 467 (Results) and 495-496 (Discussion). While we believe our results do show this difference in two independent panels (Fig 8I, N) we appreciate the qualitative nature of the result and have softened and de-emphasized the conclusion. The first statement has been changed from:

“The domain and intensity of *Ihh* OF expression in regions of incompletely fused *Hhip*^{-/-};*Pthlh*^{-/-} coronal sutures was increased compared to *Hhip*^{-/-}, although not to WT levels” to:

“The domain and intensity of *Ihh* OF expression in regions of incompletely fused *Hhip*^{-/-};*Pthlh*^{-/-} coronal sutures appeared qualitatively increased compared to *Hhip*^{-/-}, although clearly not to WT levels”.

The second statement, in the Discussion, has been deleted.

2. It would be helpful to include the lambdoid suture in the model figure to underscore the suture-specific roles for HH inhibitors.

We have provided a revised Fig 9 of the model to include the lambdoid suture, with changes made to the Figure legend as needed.

We have uploaded two versions of the manuscript, one being highlighted to indicate the text changes.

Please let me know if you think further changes are necessary.

Third decision letter

MS ID#: dev.204875R2

MS Title: Differential regulation of coronal and lambdoid suture patency by PTHLH and HHIP activity in mice

Authors: Greg Holmes; Madrikha D Saturne; Susan Motch Perrine; Qingyang Li; Joan T Richtsmeier; Ethylin Wang Jabs; Harm van Bakel

Dear Dr Holmes,

I am happy to tell you that your manuscript has been accepted for publication in Development, pending our standard publication integrity checks.